# Optimal Environmental Tax Rate in an Open Economy with Labor Migration—An E-DSGE Model Approach

Ying Tung Chan 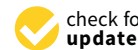

The Research Institute of Economics and Management, Southwestern University of Finance and Economics, Chengdu 611130, China; chanyt@swufe.edu.cn

**Abstract:** Recent research has started to apply environmental dynamic stochastic general equilibrium (E-DSGE) models for climate policy analysis. However, all of the studies assume a closed economy setting, where there is no interaction of the economy with an outside economy; this paper fills the gap by constructing a two-city E-DSGE model that features labor migration. With the model, we solve for the optimal environmental tax rate determined by a Ramsey social planner, who maximizes household utility and takes into account the policy's impact on labor migration. We find the following. (i) The optimal environmental tax rate should be more volatile and procyclical than the rates predicted in the aforementioned literature. (ii) In the closed economy setting, a higher environmental tax rate would always dampen production, while in our setting, it could stimulate output through deterring labor outflow and attracting labor inflow. (iii) We complement the existing literature by emphasizing that the optimal environmental tax rate in a city should respond not only to the shocks that occur internally, but also to those that occur in the opponent city. In particular, we find that it is optimal to reduce the environmental tax rate if a positive total factor productivity (TFP) shock occurs in the neighbor city.

**Keywords:** environmental tax; migration; E-DSGE model

## 1. Introduction

An elevated air pollution level has been one of the most serious problems for many of the developing countries, such as China and India. As pointed out by Huang et al. [1], 95 cities in China have their PM2.5 level exceeded 75 $\mu gm^{-3}$ for 69% of the days in January 2013, where the standard set by the United States Environmental Protection Agency (US EPA) is only 35 $\mu gm^{-3}$. Further, according to IQAir AirVisual [2], seven out of 10 most polluting cities in the world (in terms of PM2.5 level) are from India. Exposure to air pollutant, including particulate matter (PM), Ozone (O3), nitrogen dioxide (NO2), and sulfur dioxide (SO2), could trigger asthma, reduce lung function [3], and is even life-shortening [4–6]. As will be discussed in Section 2, studies show that one of the main adverse impacts of a deteriorating environment is the impediment of labor flow. Poor environmental amenity quality could deter professional workers from migrating to a city [7], and is also associated with a city's emigration rate [8–10]. Such pull and push forces could together induce a substantial loss of human capital for the economy with deteriorating environmental quality.

In this paper, we answer the question of what the optimal environmental tax rate should be if a policymaker is aware of the aforementioned brain drain (In this paper, we study the tax on ambient air pollutant in general, instead of restricting to a particular type of air pollutant. As will be discussed below, our model setting and results are applicable to the type of air pollutant once it satisfies (i) the accumulation of air pollutant in one city has an insignificant impact on those in another city; and (ii) Workers prefer to work in their home city, and labor disutility is increasing

in the level of air pollutant. Hence our model setting is more applicable in modelling local air pollutants, such as smog and particulate matter, instead of $CO_2$ and greenhouse gas that are uniformly mixed in the atmosphere.). It is worth noting that most of the existing literature on climate policy analysis (e.g., Heutel [11], Annicchiarico and Di Dio [12], Fischer and Springborn [13]) assumes a closed-economy setting and thus does not take into account the welfare impact of environmental policy on labor dynamics between cities. As labor migration is known to be one of the key factors of sustainable economic development [14,15], neglecting such an impact could result in a biased calculation of optimal environmental policy. One essential feature of climate policymaking in an open economy and stochastic environment is that the policymaking process is strategic, in the sense that the policymaker not only has to respond to the shocks that take place in his only city, but also has to take into account the shocks that occur and the decisions made by the policymaker in the neighbor city. With this feature, optimal climate policy is expected to deviate substantially from that predicted in a closed-economy setting. In this regard, we fill the gap in the literature by investigating the optimal environmental policy in an open-economy environment. In particular, we concentrate our study on computing the environmental tax rate that optimizes household welfare in the presence of stochastic elements and labor flows.

From a theoretical perspective, we extend the model of Annicchiarico and Di Dio [12] to a two-city environmental dynamic stochastic general equilibrium (E-DSGE) model, so as to clarify the underlying mechanism of how air pollutant emission and migration are related, and the corresponding macroeconomic consequences. The economic mechanism of our model is different from Annicchiarico and Di Dio [12] in at least two aspects. First, the damage function specified in Annicchiarico and Di Dio [12] is the only channel through which $CO_2$ emission deteriorates output in an economy, while in our model, output could also be affected by air pollutant emissions through the labor flow. Keeping other economic factors constant, a higher air pollutant emission level increases the labor disutility of working in the city, forces local worker to move out, and reduces the number of foreign workers that move in. Such an effect would reduce aggregate labor supply, which in turn would drive up the wage rate and deteriorate output. Second, our model is capable of explaining how the air pollutant emissions of two cities at different stages of economic development are linked. The mechanism is intuitive: if the two cities' productivity levels are asymmetric, labor would move to the city with a higher productivity, and air pollutant emissions would also diffuse from the lower- to the higher-productivity city. In sum, different from the aforementioned literature, the emissions level depend not only on shocks that occur in one city, but also on shocks in the neighbor city.

With the new model, the main result of this paper is to find the optimal tax rate under various types of shocks. Consistent with the literature (e.g., Heutel [11], Annicchiarico and Di Dio [12]), we show that the optimal environmental tax rate is procyclical, even after taking the effect of air pollutant emissions on migration into account. With a positive TFP shock, labor demand increases. More foreign workers from the opponent city would immigrate, which in turns hurts the welfare of the local household. The policymaker, whose only concern is the welfare of local households, would have an additional incentive to raise the environmental tax rate. As a result, the optimal environmental tax rate we compute should be more volatile and procyclical than the rates predicted in Annicchiarico and Di Dio [12]. Moreover, unlike in the model of Annicchiarico and Di Dio [12], where the carbon tax rate always reduces output, we show that a higher environmental tax rate could stimulate output in the long run. This is because the air pollutant emission stock, as a measure of air quality, would be reduced by the higher environmental tax rate. Better air quality, as a result, attracts more foreign workers to migrate, and output is stimulated. Further, we complement the aforementioned literature by showing that the optimal environmental tax rate should be countercyclical to the business cycles of the opponent city. That is, the Ramsey social planner would find it optimal to reduce the environmental tax rate if a positive TFP shock occurs in the opponent city. This strategy aims at preventing local workers from moving to the other city.

The remainder of this paper is structured as follows. The related literature is discussed in Section 2. Then, we present our E-DSGE model in Section 3, followed by the numerical analysis in Section 4. Finally, conclusions and potential extensions are discussed in Section 5.

## 2. Literature Review

The related literature can be categorized into three parts. Methodologically, we construct a new type of E-DSGE model for policy analysis. Hence, studies that employ an E-DSGE model for climate policy analysis will be first reviewed. Second, with labor dynamics as the main focus of the paper, we will discuss the literature on the relationship between environmental quality and labor migration. Finally, we discuss the literature on carbon leakage that study the carbon emissions linkage among countries.

### 2.1. E-DSGE Models

An E-DSGE model, a standard model framework in the macroeconomics literature, features interactions between households and firms that make dynamic and rational decisions. The economy is modeled in a stochastic environment, in that fluctuations of macroeconomic variables are caused by various exogenous shocks. One of the key features of an E-DSGE model is the consideration of intertemporal decisions of households and firms. Expecting a change in policy or exogenous factors in the future, their current decisions would respond rationally, so as to prepare for the future changes. Some studies have already emphasized the importance of intertemporal decisions in climate policy analysis (e.g., Jin [16], Jiang et al. [17]). Models without such a setting have been regarded as being subject to the Lucas critique [18].

Ours is not the first paper to apply an E-DSGE model in climate policy analysis. Heutel [11] and Fischer and Springborn [13] are the earliest works that apply such a model. Heutel [11] solves for a time-varying optimal carbon tax rate and, remarkably, shows that the optimal carbon tax rate should be procyclical. In addition, under the optimal tax rate, the economy could benefit from having dampened business cycles. On the other hand, Fischer and Springborn [13] apply a similar model to compare the properties of carbon tax rate, carbon intensity and emission cap regimes. They show that an emission cap regime is capable of reducing the volatility of most macroeconomic variables.

Subsequent studies are based on these works and have constructed more sophisticated settings. Our paper is closest to that of Annicchiarico and Di Dio [12], who emphasize the importance of the price stickiness assumption in carbon policy analysis. Notably, they find that the emission intensity regime could produce more volatile business cycles when the price level is highly rigid. It is noteworthy that the sticky price assumption is commonly viewed as necessary in analyzing the short-run behavior of macroeconomic variables. Building on Annicchiarico and Di Dio [12], who assume that the economy is closed in the sense that households and firms do not interact with the outside economy, we modify their model into a two-city model, so as to focus on the strategic interaction between the two economies through labor migrations, and to examine how such a modification could affect climate policy analysis. Their work has also been extended by Annicchiarico and Di Dio [19] and Economides and Xepapadeas [20]. The former find that the volatility of carbon emissions cycles largely depends on whether the Ramsey social planner jointly or independently determines monetary policy and the carbon tax rate. In particular, countercylical carbon emissions (in contrast with what we observe in the data) can be obtained if both policies are jointly determined. Meanwhile, with the inclusion of the energy sector, Economides and Xepapadeas [20] study how climate changes affects monetary policy formulation. They show that the nominal interest rate should respond less to an TFP shock if climate change is taken into account.

Although, so far, there is no research performing climate policy analysis in a multiple-economy setting, some of the studies do emphasize the consideration of a multisector economy. In addition to Economides and Xepapadeas [20], Dissou and Karnizova [21] extend the models of Heutel [11] and Fischer and Springborn [13] by distinguishing between energy and nonenergy sectors on the supply side. Similar to Fischer and Springborn [13], they find that the volatility of macroeconomic variables is dampened even if the emission cap is only imposed in the energy sector (Due to space

constraint, all studies that utilitize E-DSGE models are not reviewed here; other important studies include Zhang [22], Krajewski and Mackiewicz [23–25]. In addition, it is worth noting that E-DSGE model have been applied to the field of energy economics; notable examples include but are not limited to Balke and Brown [26], Aminu [27] and Mănescu and Nuño [28].).

*2.2. The Impact of Environmental Quality on Migration*

As pointed out by Gray and Bilsborrow [29], very few existing studies quantitatively estimate the relationship between environmental quality and migration (Note that there are also papers studying the effect on migration of other environmental conditions, such as rainfall [30], natural disasters [31], and temperature [32]. Further, some also examine the migration impact of air pollution on a particular group of individuals. For example, Levine et al. [33] show that air pollution could raise the chance of the migration of corporate executives.). Using U.S. data, Bayer et al. [34] estimate that on average households would pay $149 to $185 for a one-unit decline in the ambient concentrations of particulate matter, estimated by an economic model that features costly migration. Further, Gawande et al. [35] find that internal migration happens in U.S. hazardous waste sites after the individual per capita income is above a threshold, and remarkably, the threshold is approximately equal to the turning point of the EKC. Using data from 20 OECD countries, Xu and Sylwester [8] confirm that a country's Environmental Performance Index (EPI) is positively associated with its emigration rate, and the effect is more evident for the higher-educated group.

Using Chinese data, Chen et al. [10] estimate that a 10 percent increase in air pollution raises net outmigration by approximately 2.8%. They decompose the measure of migration into the net outmigration ratio and destination-based immigration ratio, and use the changes in the average strength of thermal inversions to pin down the causal relationship. To measure the monetary value of clean air in China, Freeman et al. [36] construct a residential sorting model with consideration of migration disutility. Their structural estimation shows that a unit reduction in the PM2.5 concentration was worth approximately $8.83 billion in 2005. While the data employed in both Chen et al. [10] and Freeman et al. [36] are not at the individual level, Kim and Xie [7] study a similar topic using individual-level 2015 China Census data. They show that as the number of air pollution days increases by 1, the chance of an individual moving to another province increases by 0.65% on average (In Kim and Xie [7], the day is regarded as air polluted if the air pollution index (API) is greater than 150.).

It is notable that the scale of immigration and the level of air quality are interrelated. Better air quality could attract immigrants, while on the other hand, a large inflow of labor could worsen environmental degradation (There are two strands of literature presenting opposite views on whether growth in immigration is harmful to the environment; see Ma and Hofmann [37] for a detailed discussion.). Many existing studies emphasize the correlation instead of causality of the relationship (e.g., Squalli [38,39]). However, to perform climate policy analysis with consideration of its effect on migration, the underlying mechanisms of how the two variables are related have to be carefully clarified. In this regard, Chen et al. [40] construct a dynamic model that features endogenous environmental quality and migration decisions. We differ from Chen et al. [40] in at least three aspects. (i) Our model emphasizes the strategic interaction between two economies, while only the home economy is modeled in Chen et al. [40]. (ii) We solve for the optimal environmental tax that maximizes household welfare, while no policy analysis is performed in Chen et al. [40]. (iii) Our E-DSGE model features a dynamic and rational household decision in a stochastic environment. In contrast, the household in Chen et al. [40] makes a static decision in a deterministic environment.

*2.3. Carbon Leakage*

The amount of $CO_2$ emitted by a country depends heavily on the size of its economic scale. Since the economies of two cities could be closely linked, their environmental quality could also be highly correlated. Hence, an implementation of climate policy in a (group of) countries could induce a substantial environmental impact in other countries. Such an externality impact of climate policy has

been studied extensively in the carbon leakage literature. Carbon leakage refers to the situation that the carbon emissions are mitigated in a coalition of countries that implement carbon policies, while the carbon emissions in the non-abating countries would increase. According to Antimiani et al. [41] and Babiker [42], the cross-countries diffusion of the carbon emissions is mainly through two channels: first, polluting firms switch their production plants to the non-abating countries [42]. This is also refer to as pollution haven effect [43]. Second, the decline in the energy demand in the abating countries reduces the international energy price, which stimulates the energy consumption in the non-abating countries [44,45].

Similar to the carbon leakage literature, we also study how implementing carbon taxation in a country (city) affect the carbon emission level in another city. We differ from this strand of literature in many aspects. Methodology speaking, all of the existing theoretical works studying carbon leakage use computable general equilibrium (CGE) model (e.g., [46]), instead we employ a E-DSGE model which is increasingly popular in the field of environmental economics recently. (Notice that the CGE model is more commonly used in the environmental economics literature. See Farmer et al. [47] for a detailed comparison between the CGE and E-DSGE models.). In addition to the aforementioned advantage of the model, with E-DSGE model, we are able to solve for a time-varying carbon tax rate that optimizes the social welfare.

Concerning the model mechanism, we focus on labor migration as the propagator for the diffusion of carbon emissions. With an economic shock presented or a carbon taxation implemented in a city, the economic situation of the city changes, which would initialize the labor flow. As a result, the impact of the shock is transmitted to the other city. Note that both the energy sector and the relocation of firms are not considered in our model, and hence our model mechanism is different from the aforementioned literature. Further, unlike the carbon leakage literature that the diffusion of carbon emissions is simply the outcome of asymmetric economic conditions, in our model different levels of carbon emission would in turn affect the economic conditions through changing the labor migration decision.

## 3. Model

To illustrate the impact of environmental taxation on migration, in this section, we extend the E-DSGE model of Annicchiarico and Di Dio [12] to a two-city model. Households in each city can assign their family members' labor to the city they live in or to another city. (Since the exchange rate effect is not modeled, it is emphasized that we focus on the labor dynamics between two cities, rather than those between two countries in this paper.). For simplicity, it is assumed that only labor is allowed to move across the two cities, capital does not flow freely. (Since the focus of this paper is on labor flow, we simplify the model by asssuming that capital is immobile across the cities. It is because the interaction between labor and capital flows would complicate the model mechanism. As pointed out by Antimiani et al. [41], the Global Trade Analysis Project Energy (GTAP-E) model [48,49] that used to show the pollution haven effect also does not allow for international capital mobility. Further, some studies that employ a partial-equilibrum computable general equilibrium (CGE) model to study the effect of carbon leakage do not have capital assumed in their models [50]).

Index the two cities by $i \in \{a, b\}$. In each city, assume that there is a continuum of households, indexed by $k \in [0, 1]$. The model is in discrete time, indexed by $t = 0, 1, 2, \ldots$ The households in each city are identical. The representative household in city $i$ has the following expected lifetime utility:

$$\mathbb{E}_0 \sum_{t=0}^{\infty} \beta^t \left( \ln C_{i,t} - \mu_{l,i,t} \frac{L_{ii,t}^{1+\phi}}{1+\phi} - \mu_{m,i,t} \frac{L_{ij,t}^{1+\phi}}{1+\phi} \right),$$ (1)

where $\beta \in [0, 1]$ is a discount factor. $C_{i,t}$ denotes consumption of the household living in city $i$ at time $t$. The household assigns its family members to work either in the local city or in another city. Denote as $L_{ii,t}$ and $L_{ij,t}$, respectively, the labor supply of the household living in city $i$ and working in city $i$ and $j$, where $j \neq i$. Assume that disutility is derived from the labor supply. In particular, denote as $\phi > 0$

the inverse of the Frisch elasticity. $\mu_{l,i,t}$ and $\mu_{m,i,t}$, the key parameters in our model, control the scale of the disutility of both types of labor in city $i$. It is assumed that $\mu_{l,i,t}$ is in the following form:

$$\mu_{l,i,t} = \mu_{l1} + \mu_2 M_{i,t}, \tag{2}$$

where $M_{i,t}$ is the air pollutant emission stock in city $i$ at time $t$,a and $\mu_2$ denotes the sensitivity of labor disutility to an increase in the emission stock. A higher value of $\mu_2$ implies that for the same increase in $M_{i,t}$, $\mu_{l,i,t}$ increases more, and therefore, household tends to assign more of their labor to another city. The assumption (2) is based on the discussion in Section 1 that air pollution could cause a range of health problems, and the empirical findings in Section 2 that there is a close linkage between air pollution and migration rate. $\mu_{l1}$ is the parameter that captures the disutiity of labor that is not related to the environmental concern. Similarly, $\mu_{m,i,t}$ is defined as follows:

$$\mu_{m,i,t} = \mu_{m1} + \mu_2 M_{j,t}, \tag{3}$$

where $j \neq i$. Hence, the household's migration decision is determined by the level of the emission stock in city $j$. The analogue to $\mu_{l1}$, the parameter $\mu_{m1}$ captures the disutility of migrant labor that is independent of the level of the emission stock. We set $\mu_{m1} > \mu_{l1}$ so as to capture the larger disutility generated from supplying labor in another city. The larger disutility can be caused by the transportation cost, the cost of unfamiliarity of the environment in another city and the unwillingness to leave household members. To maximize lifetime utility (1), the household is subject to the following budget constraint:

$$P_{i,t}C_{i,t} + P_{i,t}I_{i,t} + Q_{i,t}^B B_{i,t} = B_{i,t-1} + (W_{i,t}L_{ii,t} + W_{j,t}L_{ij,t}) + P_{i,t}r_{i,K,t+1}K_{i,t-1} - P_{i,t}\Gamma_I(I_{i,t}, K_{i,t-1}) + P_{i,t}D_{i,t} - T_{i,t}, \tag{4}$$

where $P_{i,t}$ is the general price level of city $i$ at time $t$. $I_{i,t}$ is the investment made by the household, living in city $i$ at time $t$. $W_{i,t}$ is the wage rate. The total labor income of the household consists of labor income from working in the home city $W_{i,t}L_{ii,t}$ and migrant city $W_{j,t}L_{ij,t}$. The household holds the amount $B_{i,t}$ in a one-period riskless bond that is issued by the local government in city $i$. The bond price is denoted as $Q_{i,t}^B$. Each unit of capital $K_{i,t-1}$ held by the household yields a rate of return $P_{i,t}r_{i,K,t+1}$, where $r_{i,K,t+1}$ is the real rate of capital return in city $i$. Adjusting the capital incurs an adjustment cost $\Gamma_I(I_{i,t}, K_{i,t-1})$, which is a function of investment and capital. Following Annicchiarico and Di Dio [12], we assume $\Gamma_K(I_{i,t}, K_{i,t}) = \gamma_I(I_{i,t}/K_{i,t} - \delta_K)^2 K_{i,t}/2$, where $\delta_K \in [0,1]$ is the depreciation rate of capital, and $\gamma_I > 0$ is a parameter that controls the scale of the cost. The convex cost function implies that the marginal adjustment cost is increasing in investment. Hence, the household tends to spread out its investment over a longer time interval. The representative household also owns all the firms in the home city. In period $t$, the household earns the amount $D_{i,t}$ of dividend from the firms. $T_{it}$ is a lump-sum tax levied by the local government in city $i$. Since capital is held by the household, the household is also subject to the law of motion of capital as follows:

$$K_{i,t} = (1 - \delta_K)K_{i,t-1} + I_{i,t}.$$

Dividing both sides of (4) by $P_{i,t}$, the budget constraint can be written as:

$$C_{i,t} + I_{i,t} + \frac{Q_{i,t}^B B_{i,t}}{P_{i,t}} = \frac{B_{i,t-1}}{P_{i,t}} + (w_{i,t}L_{ii,t} + w_{j,t}L_{ij,t}) + r_{i,K,t+1}K_{i,t-1} - \Gamma_I(I_{i,t}, K_{i,t-1}) + D_{i,t} - \frac{T_{i,t}}{P_{i,t}}, \tag{5}$$

where $w_{i,t} \equiv W_{i,t}/P_{i,t}$ and $w_{i,t} \equiv W_{j,t}/P_{i,t}$ are real wage rates. Maximizing (1) and subject to (5), gives the first-order conditions for $C_{i,t}$, $B_{i,t}$, $L_{i,t}$, $I_{i,t}$, and $K_{i,t}$ as follows:

$$C_{i,t} = 1/\lambda_{i,t} \tag{6}$$

$$R_{i,t}^{-1} = Q_{i,t}^B = \beta \mathbb{E}_t \frac{1}{\Pi_{i,t+1}} \frac{\lambda_{i,t+1}}{\lambda_{i,t}} \tag{7}$$

$$\mu_l L_{ii,t}^{\phi} = \lambda_{i,t} w_{i,t} \tag{8}$$

$$\mu_m L_{ij,t}^{\phi} = \lambda_{i,t} w_{j,t} \tag{9}$$

$$q_{i,t} = 1 + \gamma_I \left( \frac{I_{i,t}}{K_{i,t-1}} - \delta_K \right) \tag{10}$$

and

$$q_{i,t} = \beta \mathbb{E}_t \frac{\lambda_{t+1}}{\lambda_t} \left[ r_{i,K,t+1} + \frac{\gamma_I}{2} \left( \frac{I_{i,t+1}}{K_{i,t}} - \delta_K \right)^2 + \gamma_I \left( \frac{I_{i,t+1}}{K_{i,t}} - \delta_K \right) \frac{I_{i,t+1}}{K_{i,t}} + (1 - \delta_K) q_{i,t+1} \right], \tag{11}$$

where $\lambda_{i,t}$ is a Lagrangian multiplier of city $i$ at time $t$. $R_{i,t}$ is the gross interest rate of city $i$ in period $t$. $\Pi_{i,t}$ denotes the gross inflation rate in city $i$ at time $t$. Combining (6) and (7) yields the familiar Euler equation $R_{i,t}^{-1} = \beta \mathbb{E}_t[C_{i,t+1}/(C_{i,t}\Pi_{i,t+1})]$ that determines the household's intertemporal tradeoff of consumption. The conditions (8) and (9) are the labor supply of the household in city $i$ and city $j$, respectively. It is evident that the labor supplies are increasing in the wage rate of the distinction city. Since capital is not allowed to flow freely across the cities, the households in the two cities invest independently. $q_{i,t}$ in (10) is the so-called Tobin's q, which is the capital price (per unit of consumption). It equals the per unit price of the investment good (which is 1) plus the marginal change in the adjustment cost. In the absence of the capital adjustment cost ($\gamma_I = 0$), we have $q_{i,t} = 1$. That is, the capital price is identical to the per unit price of the investment good. Further, Equation (11) states that the capital price is also equal to the sum of the current rate of return to capital and the future capital price, discounted by the depreciation rate.

### 3.1. Firms

The setting of the firm problem is standard. Denote as $Y_{i,t}$ the aggregate output of city $i$, and $Y_{i,t}(h)$ the corresponding intermediate goods production. It is assumed that the final good producers are symmetric and act under perfect competition. The intermediate goods sector consists of a continuum of monopolistically competitive polluting producers. In particular, $Y_{i,t}$ is a composite of a continuum of intermediate goods $Y_{i,t}(h)$, where $h \in [0,1]$ and $i \in \{a,b\}$ according to a constant elasticity of substitution (CES) function as follows:

$$Y_{i,t} = \left( \int_0^1 Y_{i,t}(h)^{\frac{\theta-1}{\theta}} dh \right)^{\frac{\theta}{\theta-1}},$$

where $\theta > 0$ controls the substitutability between any two intermediate goods. The higher the $\theta$, the more substitutable the goods are. This leads to a downward-sloping demand function for intermediate good $i$ as follows:

$$Y_{i,t}(h) = \left( \frac{P_{i,t}(h)}{P_{i,t}} \right)^{-\theta} Y_{i,t} \tag{12}$$

where $P_{i,t}$ and $P_{i,t}(h)$ are, respectively, the aggregate price and the price of intermediate good $h$ in city $i$. Using the budget constraint $P_{i,t}Y_{i,t} = \int_0^1 P_{i,t}(h)Y_{i,t}(h)dh$, it is straightforward to compute the aggregate price level as $P_{i,t} = (\int_0^1 P_{i,t}(h)^{1-\theta}dh)^{1/(1-\theta)}$.

Assume that producing an intermediate good $h$ in city $i$ requires both labor $L_{i,t}(h)$ and capital $K_{i,t-1}(h)$. Both the labor and capital markets are assumed to be perfectly competitive. We set the production function to be Cobb-Douglas as follows:

$$Y_{i,t}(h) = (1 - Y(M_{i,t})) A_{i,t} L_{i,t}(h)^{1-\alpha} K_{i,t-1}(h)^{\alpha}. \tag{13}$$

In (13), $\alpha \in [0,1]$ is the share of capital to output. Note that from the firms' perspective, there is no distinction between the migrant labor and domestic labor. Apart from the standard Cobb–Douglas production function, the additional term $(1 - Y(M_{i,t}))$ in (13) refers to the negative externality arise from environmental degradation. $M_{i,t}$ is the aggregate air pollutant emission stock of city $i$ at time $t$. It is emphasized that $M_{i,t}$ is common to all the firms and thus is without the firm index $h$. $Y(M_{i,t})$, whose range is from 0 to 1, is a damage function that measures the percentage reduction in output owing to the air pollutant emission accumulated. $A_{i,t}$ is the level of TFP in city $i$ at time $t$. It is assumed that the logarithm of the TFP levels is random and follows an AR(1) process as

$$\ln(A_{i,t}/A) = \rho_A \ln(A_{i,t-1}/A) + \varepsilon_{A_i,t}, \tag{14}$$

where $\rho_A \in [0,1]$ and $A$ denote, respectively, the persistence and the steady-state value of the processes. $\varepsilon_{A_i,t}$ is white noise that is identically independently distributed, and follows a normal distribution $N(0, \sigma_A)$, where $\sigma_A > 0$ is the standard deviation of $\varepsilon_{A_i,t}$. For simplicity, we set the parameters $\{\rho_A, A, \sigma_A\}$ to be the same across the cities, while noting that the generalization to heterogeneous processes is straightforward.

Air pollutant emission $Z_{i,t}(h)$, as a byproduct of output, is modeled as follows:

$$Z_{i,t}(h) = \varphi_{i,t}(1 - U_{i,t}(h))Y_{i,t}(h) \tag{15}$$

where $U_{i,t}(h)$ is an abatement effort exerted by firm $h$ in city $i$ at time $t$. It is assumed that the air pollutant emission is a linear function of output, and hence, the abatement effort $U_{i,t}(h)$ is indeed a term that controls for the air pollutant intensity level. $\varphi_{i,t} > 0$ is a scale parameter that determines the amount of air pollutant emission per unit of output in city $i$. It can also be considered as the average abatement technology or abatement equipment levels in the city. By setting $U_{i,t}(h) = 0$, it can be seen that $\varphi_{i,t}$ measures emissions per unit of output in the absence of abatement effort. Indeed, rewriting (15) into the form $(Z_{i,t}(h)/Y_{i,t}(h))/\varphi_{i,t} - 1 = -U_{i,t}(h)$ reveals that the abatement effort is indeed the percentage reduction of air pollutant intensity from the benchmark intensity level where $U_{i,t}(h) = 0$.

Note that the TFP shock introduced above would simultaneously affect the emission level and households' migration decision. To examine how households' migration decisions respond to an exogenous increase in the air pollutant emission level, it is necessary to introduce a shock process to the abatement inefficiency parameter $\varphi_{it}$. Similarly, it is assumed that the logarithm of $\varphi_{i,t}$ follows an AR(1) process as follows:

$$\log(\varphi_{i,t}/\varphi) = \rho_{\varphi_i} \log(\varphi_{i,t-1}/\varphi) + \varepsilon_{\varphi_i,t}, \tag{16}$$

where $\rho_{\varphi_i} \in [0,1]$ and $\varphi$ are the persistence and the steady-state values of the process, respectively. $\varepsilon_{\varphi_i,t}$ is white noise that is normally distributed with mean 0 and standard deviation $\sigma_{\varphi_i}$.

Following Annicchiarico and Di Dio [12], the abatement cost function for firm $h$ in city $i$ at time $t$ is

$$C_{A,i,t}(h) = \phi_1 U_{i,t}(h)^{\phi_2} Y_{i,t}(h), \tag{17}$$

where $\phi_1 > 0$ is a scale parameter, and $\phi_2 > 1$ is the key parameter that controls the curvature of the abatement cost to the abatement effort. With $\phi_2 > 1$, the marginal abatement cost to abatement effort $\partial C_{A,i,t}(h)/\partial U_{i,t}(h) = \phi_1\phi_2 U_{i,t}(h)^{\phi_2-1}Y_{i,t}(h)$ is still increasing in $U_{i,t}(h)$. Such a convexity assumption incentivizes firms to separate the abatement process into several periods.

The instantaneous profit function of firm $h$ in city $i$ at time $t$ is written as follows:

$$\pi_{i,t}(h) = \max_{L_{i,t}(h),K_{i,t-1}(h),U_{i,t}(h)} P_{i,t}(h)Y_{i,t}(h) - W_{i,t}L_{i,t}(h) - R_{K,i,t}K_{i,t-1}(h) - C_{A,i,t}(h) - P_{Z,i,t}Z_{i,t}(h). \tag{18}$$

The firm maximizes the above profit function by choosing labor $L_{i,t}(h)$, capital $K_{i,t-1}(h)$, and abatement effort $U_{i,t}(h)$, subject to the constraints (12), (13), (15) and (17). As stated in (18), the only

source of firms' income comes from the revenue $P_{i,t}(h)Y_{i,t}(h)$, while the total variable cost includes labor cost $W_{i,t}L_{i,t}(h)$, capital rental cost $R_{K,i,t}K_{i,t-1}(h)$, abatement cost $C_{A,i,t}(h)$, and environmental tax payment $P_{Z,i,t}Z_{i,t}(h)$. The environmental tax rate $P_{Z,i,t}$ is assumed to be different across the cities. The first-order conditions for $K_{i,t-1}(h)$, $L_{i,t}(h)$, and $U_{i,t}(h)$ are, respectively,

$$\alpha \frac{Y_{i,t}(h)}{K_{i,t-1}(h)} \Psi_{i,t} = \frac{R_{K,i,t}}{P_{i,t}} \equiv r_{K,i,t} \tag{19}$$

$$(1-\alpha) \frac{Y_{i,t}(h)}{L_{i,t}(h)} \Psi_{i,t} = \frac{W_{i,t}}{P_{i,t}} \equiv w_{i,t} \tag{20}$$

and

$$\varphi p_{z,i} = \phi_1 \phi_2 U_{i,t}(h)^{\phi_2 - 1}, \tag{21}$$

where $r_{K,i,t}$ and $w_{i,t}$ are the real capital return and real wage rate in city $i$ at time $t$. According to (19) and (20), the optimal capital and labor are determined simply by equalizing the marginal product of capital and marginal product of labor to the real rate of capital and real wage rate, respectively. Note that the labor supply curves for domestic and migrant workers are different, due to the difference in the degree of labor disutility. However, both face the same labor demand curve (20) since the labor they supply is homogeneous. The $p_{z,i,t} = P_{Z,i,t}/P_{i,t}$ is the real environmental tax rate in city $i$ at time $t$. In (19) and (20), $\Psi_{i,t}$ is the marginal cost of production that arose from capital and labor, and is

$$\Psi_{i,t} = \frac{1}{\alpha^\alpha (1-\alpha)^{1-\alpha} A_{i,t}(1 - Y(M_{i,t}))} w_{i,t}^{1-\alpha} r_{K,i,t}^\alpha. \tag{22}$$

By taking the abatement cost and environmental tax payment into account, the total marginal cost equals

$$MC_{i,t}(h) = \Psi_{i,t} + \phi_1 U_{i,t}(h)^{\phi_2} + \frac{P_{Z,i,t}}{P_{i,t}}(1 - U_{i,t}(h))\varphi. \tag{23}$$

### 3.2. Nominal Rigidity

As emphasized by Annicchiarico and Di Dio [12], the price stickiness assumption is crucial in determining the effectiveness of environmental policies. In this regard, we follow them to impose the sticky-price assumption à la Calvo [51] in our model. In particular, it is assumed that there is a $\nu \in [1,0]$ portion of firms that are restricted in fixing their price level in each period. The remaining $1 - \nu$ portion of firms can reset their new prices. Put differently, each firm has a probability $\nu$ of adjusting its price in each period. Assume that the chance of price adjustment is independent across the firms. Denote as $P_{i,t}^*(h)$ the new price set by firm $h$ located in city $i$ at time $t$. The firm chooses $P_{i,t}^*(h)$ to maximize its expected lifetime profit as

$$\max_{P_{i,t}^*(i)} \mathbb{E}_t \sum_{k=0}^{\infty} \nu^k Q_{t,t+k} \left[ P_{i,t}^*(h)Y_{i,t+k}(h) - MC_{i,t}(h)Y_{i,t+k}(h) \right], \tag{24}$$

subject to the demand function (12). In (24), $Q_{t,t+k,i} = \beta^k (\lambda_{t+k,i}/\lambda_{t,i})$ is the stochastic discount factor between time $t$ and time $t + k$ in city $i$. The first-order condition for $P_{i,t}^*(h)$ satisfies

$$p_t^* \equiv \frac{P_t^*}{P_t} = \mathcal{M} \frac{\mathbb{E}_t \sum_{k=0}^{\infty} \nu^k Q_{t,t+k,i} MC_{i,t+k} \left(\frac{P_{t+k}}{P_{i,t}}\right)^\theta Y_{i,t+k}}{\mathbb{E}_t \sum_{k=0}^{\infty} \nu^k Q_{t,t+k,i} \left(\frac{P_{i,t+k}}{P_{i,t}}\right)^\theta Y_{i,t+k}}. \tag{25}$$

Since all the firms in the same city that have a chance to adjust their prices would make the same decision, the firm index $h$ is skipped in Equation (25). $\mathcal{M} \equiv \theta/(\theta - 1) > 1$ is the markup when

the price is perfectly flexible. To see this, note that Equation (25) converges to $P_{i,t}(h) = \mathcal{M}MC_{i,t}(h)$ as $v \to 0$. Since $\mathcal{M} > 1$, the firm would charge a price above its marginal cost of production, revealing that the firm is able to earn a profit. The lower the value of $\theta$ is, the higher is the value of $\mathcal{M}$ and the less profit the firm can make.

### 3.3. Public Sector and Nominal Interest Rate

Note that business cycles and air pollutant emissions in a city can also be affected by policies drawn up by the local government (In this paper, it is assumed that the regional government in both cities make decision separately, and do not strategically interact. This simplification is made because both fiscal and monetary policies are not the focus of this paper.). In the public sector, it is assumed that the government budget is balanced in both cities in each period. That is,

$$T_{i,t} + p_{Z,i}Z_{i,t} = G_i, \tag{26}$$

where $G_i$ is the real expenditure of the local government in city $i$, and is assumed to be exogenous and constant over time. The left-hand side of the equation states that government income is the sum of real lump-sum tax revenue $T_{i,t}$ and real environmental tax revenue $p_{Z,i}Z_{i,t}$.

The local government in city $i$ issues one-period riskless bonds with return $R_{i,t}$. By controlling the bond supply, the bond return can be manipulated. In particular, we follow the literature in assuming that the local government decides the bond return according to the Taylor rule as

$$R_{i,t} = R \left( \frac{\Pi_{i,t}}{\Pi} \right)^{\iota_\pi} \left( \frac{Y_{i,t}}{Y} \right)^{\iota_Y}, \tag{27}$$

where $\iota_\pi > 0$ and $\iota_Y > 0$ are the elasticity of the interest rate with respect to the gross inflation rate $\Pi_{i,t}$ and output level $Y_{i,t}$. In other words, the values of $\iota_\pi$ and $\iota_Y$ determine the sensitivity of $R_{i,t}$ to the change in inflation and output, respectively. A higher value of $\iota_\pi$ ($\iota_Y$) implies that the interest rate would rise more per unit change in the inflation rate (output). $\Pi$ and $Y$ are the steady-state values of the gross inflation rate and output, respectively.

### 3.4. Equilibrium

With the Calvo pricing setting, the aggregate price level is $P_{i,t} = [v P_{i,t-1}^{1-\theta} + (1-v) P_{i,t}^{*1-\theta}]^{1/(1-\theta)}$, where $P_{i,t}^*$, specified in Equation (25), is the new price level set by the firms in city $i$ who can adjust their price level in period $t$. That is, the aggregate price level in city $i$ is the weighted average of the previous price level and the new price level set by the firms. Dividing both sides of the formula by $P_{i,t}$ yields $1 = [v \Pi_{i,t-1}^{\theta-1} + (1-v) p_{i,t}^{*1-\theta}]^{1/(1-\theta)}$, where $\Pi_{i,t}^* \equiv P_{i,t}/P_{i,t-1}$ is the gross inflation rate and $p_{i,t}^* \equiv P_{i,t}^*/P_{i,t}$.

In equilibrium, firms in the same city would exert the same amount of abatement effort that satisfies Equation (21), that is, $U_{i,t}(h) = U_{i,t}$. In contrast, capital and labor demand is not the same across the firms due to the price stickiness assumption. Let $K_{i,t-1}$ and $L_{i,t}$ be the aggregate capital and labor of city $i$ at time $t$. We have $K_{i,t-1} = \int_0^1 K_{i,t-1}(h)dh$ and $L_{i,t} = \int_0^1 L_{i,t}(h)dh$. Aggregating firms' output using the demand function (12) yields $\int_0^1 Y_{i,t}(h)dh = \int_0^1 (P_{i,t}(h)/P_{i,t})^{-\theta} Y_{i,t}dh = D_{i,p,t}Y_{i,t}$, where $D_{i,p,t} \equiv \int_0^1 (P_{i,t}(h)/P_{i,t})^{-\theta}dh$ is a measure of price dispersion in city $i$ at time $t$, and is bounded below by one [52]. According to the price adjustment mechanism, $D_{i,p,t}$ evolves according to the first-difference equation $D_{i,p,t} = (1-v)p_{i,t}^{*-\theta} + v \Pi_{i,t}^\theta D_{i,p,t-1}$. The aggregate capital and labor demand functions can be found by integrating both sides of Equations (19) and (20) with respect to $h$. That is, $D_{i,p,t}Y_{i,t} = \int_0^1 Y_{i,t}(h)dh = r_{i,K,t}K_{i,t-1}/(\Psi_{i,t}\alpha)$ and $D_{i,p,t}Y_{i,t} = \int_0^1 Y_{i,t}(h)dh = w_{i,t}L_{i,t}/(\Psi_{i,t}(1-\alpha))$. Indeed, the production function in terms of aggregate output $Y_{i,t}$ can be pinned down using the aggregate capital and labor demand function, and we have $Y_{i,t} = (1 - \Upsilon(M_{i,t}))A_{i,t}K_{i,t-1}^\alpha L_{i,t}^{1-\alpha}D_{i,p,t}^{-1}$. Comparing (13) with the aggregate production function, the existence of price dispersion reduces

the aggregate output. Note that $L_{i,t}$ is the equilibrium labor employed in city $i$, which is the sum of the domestic labor and migrant labor:

$$L_{i,t} = L_{ii,t} + L_{ji,t}, \tag{28}$$

for $j \neq i$ (Condition (28) implicitly assumes that the labor supplied by households in both cities is homogenous. It is straightforward to capture labor heterogeneity by generalizing the condition into $L_{i,t} = L_{ii,t} + \kappa L_{ji,t}$, where $\kappa > 0$ is the labor productivity supplied by households in city $i$ relative to those in city $j$.). In a good market equilibrium, the aggregate output $Y_{i,t}$ should equal the sum of good demand in the city. This obtains the following:

$$Y_{i,t} - w_{i,t}L_{ji,t} + w_{j,t}L_{ij,t} = C_{i,t} + I_{i,t} + G_i + C_{A,i,t} + \frac{\gamma_I}{2}\left(\frac{I_{i,t}}{K_{i,t-1}} - \delta_K\right)^2 K_{i,t-1}. \tag{29}$$

The GDP accounting Equation (29) states that the aggregate good demand in city $i$ at time $t$ consists of consumption $C_{i,t}$, investment $I_{i,t}$, government expenditure $G_i$, aggregate abatement cost $C_{A,i,t}$ and capital adjustment cost $\gamma_I/2(I_t/K_t - \delta_K)^2K_t$, where $C_{A,i,t} = \int_0^1 C_{A,i,t}(h)dh = \phi_1 U_{i,t}^{\phi_2}Y_{i,t}D_{i,p,t}$. Since not all the output is produced by local workers, on the left-hand-side of (29) $Y_{i,t}$ is subtracted by the (real) income earned by the foreign workers $w_{i,t}L_{ji,t}$ and added by the (real) income earned by local worker from the foreign country $w_{j,t}L_{ij,t}$.

Finally, the aggregate emitted air pollutant $Z_{i,t}$ in city $i$ at time $t$ can be found by the formula $Z_{i,t} = \varphi(1 - U_{i,t})Y_{i,t}D_{i,p,t}$. The emission stock is accumulated according to the following equation.

$$M_{i,t} = (1 - \delta_M)M_{i,t-1} + Z_{i,t} + Z_{i,t}^*, \tag{30}$$

where $\delta_M \in [0,1]$ is the depreciation rate of the emission stock. $Z_{i,t}^*$ is the foreign air pollutant emissions that help accumulate the emission stock of city $i$ at time $t$.

### 3.5. Mitigation

Note that when a shock occurs, households would change the amount of labor assigned in both cities. Hence, to measure whether a shock would increase the chance of a household migrating, it is more accurate to use the ratio of migrated workers to domestic workers as a measure of the degree of migration. In particular, we define

$$Lr_{i,t} \equiv \frac{L_{ij,t}}{L_{ii,t}}, \tag{31}$$

be the ratio of labor migrated from city $i$ to city $j$ to nonmigrated labor in city $i$ at time $t$. A higher value of $Lr_{i,t}$ indicates that households in city $i$ assign more labor to city $j$ per unit of domestic labor, which could represent an increase in the household's intention to migrate. In addition, we denote with $\tilde{L}_{i,t}$ the total labor supplied by the household in city $i$, that is:

$$\tilde{L}_{i,t} \equiv L_{ii,t} + L_{ij,t}. \tag{32}$$

## 4. Quantitative Analysis

### 4.1. Choice of Parameters

The choice of parameters is mostly based on the literature (All the parameters used directly follow the choice in Annicchiarico and Di Dio [12] with two exceptions: (i) the parameters of the two shock processes are set to be equal. First, the choice of these parameters would not affect the long-run analysis in Section 4.2 where the shocks are absent. Second, they only affect the persistence of the impulse response functions in Section 4.3. Since our results are qualitative, they are independent of these parameters. (ii) Parameters of labor disutility $\mu_{l,i}$, $\mu_{m,i}$, and $\mu_2$ are new to this paper.

Again, our qualitative result only relies on the condition where $\mu_{m,i} > \mu_{l,i}$. Further, various values of $\mu_2$ are considered in Figure 2 to ensure that the results are robust.) For simplicity, the two cities are assumed to be symmetric in the sense that most of the parameters in the two cities are assumed to be the same. The model is set so that each period represents a quarter.

On the firm side, it is assumed that the Calvo pricing parameter $\nu$ is 3/4, implying that only one-fourth of firms can adjust their price every quarter. The elasticity of substitution $\theta$ is set to 6, which yields a 20% $(6/(6-5) = 1.2)$ markup for the intermediate good firms when the price is flexible. The share of capital to output in the production function $\alpha$ is set to 1/3. Further, we set the rate of depreciation of capital $\delta_K = 0.025$, which translates into a 10% $(2.5\% \times 4)$ annual depreciation rate of capital.

For the parameters concerning the household problem, we set the discount factor $\beta$ to 0.99, which implies a (quarterly) discount rate of 1% $(1/0.99 - 1 \approx 1\%)$. For labor disutility, the inverse of the Frisch elasticity $\phi$ is set to 1. As discussed below, the steady-state value of emission stock $M_i$ is around 800, and hence, we simply set $\mu_2$ to 1/800 (As pointed out by Annicchiarico and Di Dio [12], the U.S. carbon mass is approximately 800 gigatons in 2005). In addition, we set the scale parameter $\mu_{l,i} = 2 < 2.5 = \mu_{m,i}$, so households have to pay a higher cost of working in the foreign city. The function form of the capital adjustment cost and the corresponding parameters follow Christiano et al. [53], so we have $\gamma_I = 0.5882$.

Concerning the setting of the environmental elements, we mainly based our choice of parameters on Annicchiarico and Di Dio [12]. Specifically, we assume that the rate of depreciation of emission stock $\delta_M$ is 0.0021, which is equivalent to a depreciation rate of 0.8% $(0.0021 \times 4 \approx -0.8\%)$ annually. Moreover, the steady-state value $\varphi$ in (16) is set to 0.45. Following Annicchiarico and Di Dio [12], the damage function is set to be quadratic as $Y(M_{i,t}) = \gamma_0 + \gamma_1 M_{i,t} + \gamma_2 M_{i,t}^2$ where $(\gamma_0, \gamma_1, \gamma_2) = (1.395e^{-3}, -6.6722e^{-6}, 1.4647e^{-8})$. The foreign emission level $Z^*$ is chosen such that the emission stock equals 800. It turns out that $Z^* = 0.745$. For the abatement cost function, we follow Annicchiarico and Di Dio [12] to set $\phi_1 = 0.185$ and $\phi_2 = 2.8$. The high value of $\phi_2$ implies that the abatement cost is sensitive to a per-unit increase in the abatement effort. Hence, firms would prefer to separate the abatement process into multiple periods.

In the public sector, following the calibration procedure of Annicchiarico and Di Dio [12], the value of government expenditure in both cities $G_i$ is found, so the government expenditure to output ratio $G_i/Y_i = 0.22$. For monetary policy, we follow the convention of the macroeconomic literature to set the elasticity parameters $\tau_\pi$ and $\tau_Y$ to 3 and 0.25, respectively.

Finally, the parameters of the TFP shock process (14) are also adopted from Annicchiarico and Di Dio [12], such that $\rho_{A,i} = 0.95$ and $\sigma_{A,t} = 0.0045$ for all city $i$. For a fair comparison, the corresponding parameters of the abatement inefficiency shocks are also set to be the same as those of the TFP shock. That is, we set $\rho_{\varphi,i} = 0.95$, and $\sigma_{\varphi,i} = 0.0045$ for all the city $i$. Finally, the steady-state value of $A_i$ is assumed to be 1.

Table 1 summarizes the parameter values discussed above, and the first-order conditions of the model are displayed in Appendix A. With the parameter values specified, the deterministic steady-state values of the endogenous variables can be solved by assuming the stocks are absent. Then, the model is solved by applying the first-order perturbation method around the deterministic steady state. We separate the numerical exercises into short-run and long-run analysis. For the long-run analysis, Section 4.2 presents how the steady-state values of air pollutant emissions and labor supply respond to the steady-state values of the shocks and environmental tax rate. For the short-run analysis, Section 4.3 answers a similar question by presenting the corresponding impulse response functions (IRFs) of air pollutant emissions and labor supply.

**Table 1.** The calibrated parameter values used for numerical analysis.

| Parameters | Value | Description |
|:---:|:---:|:---:|
| $\beta$ | 0.99 | Discount factor |
| $\phi$ | 1 | Inverse of Frisch elasticity |
| $\mu_{l,i}$ | 2 | Parameter of labor disutility |
| $\mu_{m,i}$ | 2.5 | Parameter of labor disutility |
| $\mu_2$ | 1/800 | Parameter of labor disutility |
| $\delta_K$ | 0.025 | Depreciation rate of capital |
| $\gamma_I$ | 0.5882 | Parameter of capital adjustment cost |
| $\theta$ | 6 | Elasticity of substitution between intermediate goods |
| $\nu$ | 0.75 | Parameter of Calvo pricing adjustment |
| $\alpha$ | 1/3 | Share of capital in production |
| $\phi_1$ | 0.185 | Parameter of abatement cost |
| $\phi_2$ | 2.8 | Parameter of abatement cost |
| $\delta_M$ | 0.0021 | Depreciation rate of emission stock |
| $Z^*$ | 1.3299 | Foreign emission level |
| $\varphi$ | 0.45 | Marginal emission of production |
| $\gamma_0$ | $1.395e^{-3}$ | Parameter of damage function |
| $\gamma_1$ | $-6.6722e^{-6}$ | Parameter of damage function |
| $\gamma_2$ | $1.4647e^{-8}$ | Parameter of damage function |
| $\tau_\pi$ | 3 | Parameter of inflation gap |
| $\tau_Y$ | 0.25 | Parameter of output |
| $A_i$ | 1 | Steady-state value of technology level of city $i$ |
| $p_{Z,i}$ | 0.05 | environmental tax rate of city $i$ |
| $\rho_{A,i}$ | 0.95 | Persistence of technology shock of city $i$ |
| $\rho_{\varphi,i}$ | 0.95 | Persistence of abatement inefficiency shock of city $i$ |
| $\sigma_{A,i}$ | 0.0045 | Standard deviation of technology shock of city $i$ |
| $\sigma_{\varphi,i}$ | 0.0045 | Standard deviation of abatement inefficiency shock of city $i$ |

*4.2. Long-Run Analysis*

4.2.1. The Impact of Emission Level on Migration

To examine the impact of emission on households' migration decision, we compute the values of endogenous variables in the deterministic steady-state with different steady-state values of $\varphi_a$. A higher value of $\varphi_a$ implies that more air pollutant is emitted per unit of production in city $a$, keeping constant all the other macroeconomic factors, such as TFP level, government expenditure, and monetary policy in both cities.

Figure 1 displays the steady-state values of labor and air pollutant emission against $\varphi_a$ (The value of $\varphi_b$ is assumed to be constant and equals 0.45). First, it is intuitive that an increase in $\varphi_a$ leads to more emission $Z_a$ in city $a$. This in turns results in a larger emission stock $M_a$, and thus discourages households in both cities from working in city $a$. Therefore, both $L_{aa}$ and $L_{ba}$ drop accordingly. The household in city $a$ would like to assign more of its labor to city $b$, thus resulting in an increasing $L_{ab}$ against $\varphi_a$. In equilibrium, the labor worked in city $a$ ($L_a$) decreases, while the labor worked in city $b$ ($L_b$) increases. Hence, firms in city $b$ benefit from the growth of the labor supply, and would therefore expand their production. Eventually, the emission level in city $b$ ($Z_b$) also increases. However, without any change in productivity in both cities, the higher labor supply in city $b$ would drive down the wage rate in the city, and discourage local workers in city $b$ from working. The increasing labor ratio $Lr_a$ in Figure 1 reveals that the household in city $a$ would assign more labor in city $b$ per unit labor supply in city $a$. Likewise, the decreasing labor ratio $Lr_b$ also indicates that the household in city $b$ also prefers to stay in city $b$.

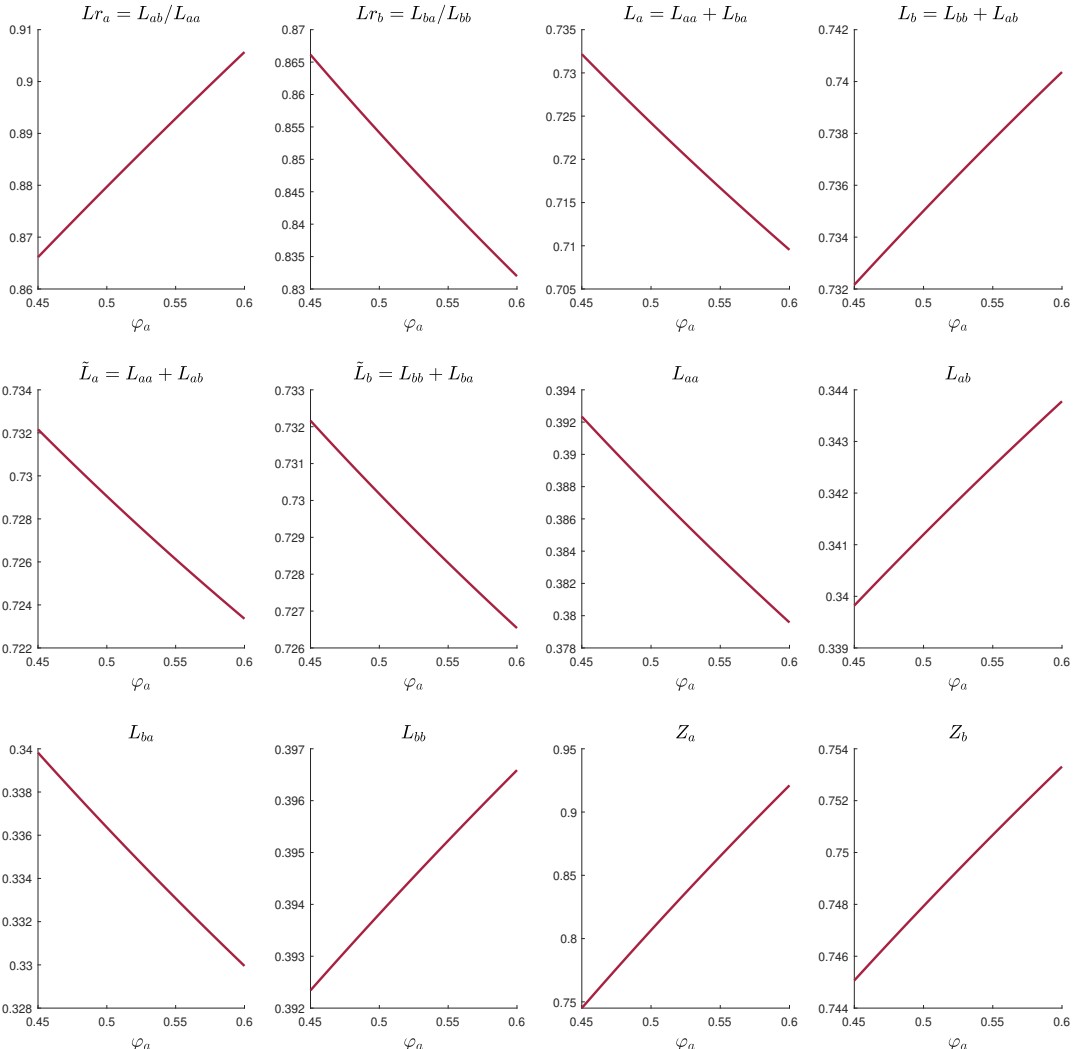

**Figure 1.** Steady-state values of labor and air pollutant emission levels in cities *a* and *b* against $\varphi_a$.

Overall, we show that keeping macroeconomic factors unchanged, more emissions in a city would deter workers from moving in to the city, while pushing the local workers move out. The increase in $\varphi_a$ produces a negative relationship between the migration and emission level, which is consistent with and provides a theoretical foundation for the empirical results discussed in Section 2. Further, although the two cities could be far away from each other, an increase in a city's air pollutant emission would increase the emission level of the opponent city through the labor migration process.

### 4.2.2. Total Factor Productivity

Since an advance in a city's productivity level could simultaneously lead to higher output and air pollutant emission, immigrants are, on one hand, attracted by the higher output level but, on the other hand, deterred by the polluted environment. To understand the tradeoff between these two forces, the long-term impact of economic advancement on migration and air pollution is presented in this subsection. In particular, we compute the values of endogenous variables in the deterministic steady-state with different steady-state value of $A_a$. Figure 2 plots the steady-state values of labor and emission level in both cities against $A_a$ with the environmental concern $\mu_2 = 0, 1/800$, and $2/800$.

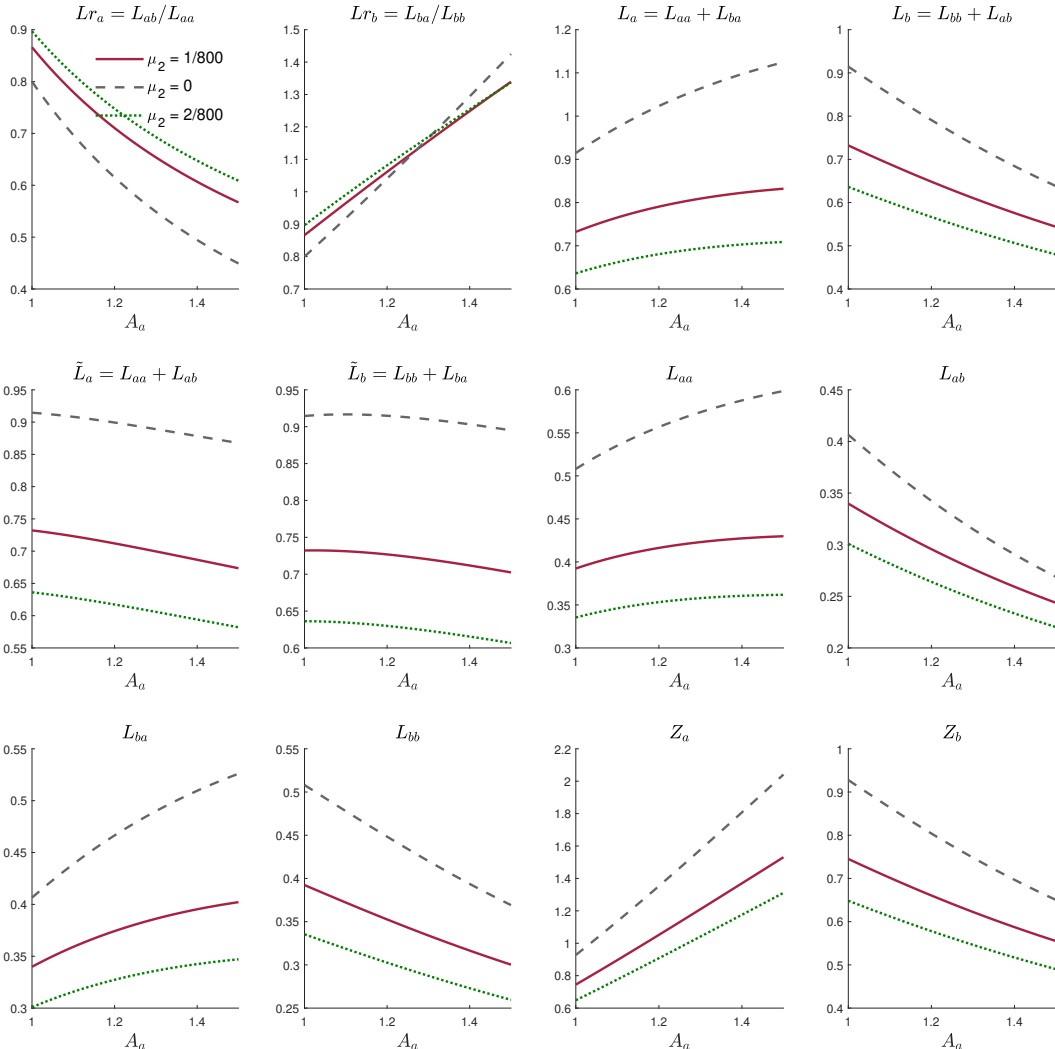

**Figure 2.** Steady-state values of labor and air pollutant emission levels in cities *a* and *b* against the TFP level $A_a$. The solid, dashed and dotted lines represent the models with $\mu_2 = 1/800, 0$, and $2/800$, respectively.

First, it is clear that higher TFP in city *a* attracts more labor to the city and hence increases the equilibrium labor $L_a$ in city *a*, while reducing the equilibrium labor $L_b$ in city *b*. It is worth noting that $L_{aa}$ is increasing and $L_{ab}$ is decreasing against $A_a$, which implies that the household in city *a* switches labor to its home city as the home productivity level improves. This is because higher TFP raises the productivity of firms located in city *a* and thus leads to higher marginal productivity of labor and capital. As a result, firms would hire more labor and are willing to offer a higher wage rate. Likewise, the higher wage rate attracts households in city *b* to migrate to the city, and encourages them to switch their labor supply from city *b* to city *a*, which thus leads to an increasing $L_{ba}$ and a decreasing $L_{bb}$ against $A_a$. In sum, as $A_a$ increases, the higher level of $L_a$ is indeed contributed to by both the local workers $L_{aa}$ from city *a* and the migrant workers $L_{ba}$ from city *b*. In contrast, since both $L_{ab}$ and $L_{bb}$ decrease, total labor in city *b* drops. The decreasing labor ratio $Lr_a$ and the increasing labor ratio $Lr_b$ both indicate that households in both cities are more likely to assign their labor to city *a*. Put differently, higher productivity in a city raises the chance of foreign workers moving in, and simultaneously reduces the chance of domestic workers moving out. Since more economic activities are concentrated in city *a*, it is evident that the emission $Z_a$ would increase, while the emission $Z_b$ falls. To conclude, higher productivity in a city would result in a positive relationship between emission level $Z_a$ and degree of migration $Lr_b$. In addition, the emission levels in the two cities are negatively correlated.

Next, we compare the path of variables with $\mu_2 = 0$ and $1/800$. By increasing the value of $\mu_2$, we are able to examine the role of environmental concern on households' migration decision. As shown in Figure 2, a higher value of $\mu_2$ reduces both domestic and migrant labor in both cities. This is because, according to (2) and (3), an increase in $\mu_2$ raises both $\mu_{l,i,t}$ and $\mu_{m,i,t}$, making both types of labor supply more undesirable to the households. Hence, aggregate labor in city $a$ $L_a$ and city $b$ $L_b$ drop simultaneously, while it is notable that the extent of changes in the two types of labor are different. As seen above, the labor ratio $Lr_a$ is increasing with $\mu_2$, indicating that household $a$ tends to assign relatively more of its labor in city $b$. This is intuitive, since workers who work in foreign city have a higher labor cost than those who work locally, according to the assumption that $\mu_{m,i} > \mu_{l,i}$. With a higher environment cost $\mu_2$, the higher fixed cost incurred by migration becomes less relevant, making households more willing to work in the foreign city.

### 4.2.3. Environmental Taxation

Regarding the adverse impact of air pollutant emission on migration, one might wonder whether raising the environmental tax rate can boost output through reducing the air pollutant emission level and attracting labor inflow. Figure 3 plots the deterministic steady-state values of labor and air pollutant emissions against different values of environmental tax rate $p_{Za}$ in city $a$. The environmental tax rate in city $b$ is assumed to be 0.05. First, note that the labor ratio $Lr_a$ exhibits a U shape against $p_{Za}$. That is, a household in city $a$ is less likely to migrate when the tax rate is slightly above 0, while the chance of migration increases as the tax rate increases. The mechanism of the environmental tax rate in mitigating air pollutant emission is through raising the firms' cost of environmental tax payment per unit of emission and thus forcing the firms to exert more effort on abatement. However, according to (23), firms in city $a$ would face a higher marginal cost of production under a higher environmental tax rate, and as a result, they tend to reduce their scale of production and hire less labor. Hence, on the one hand, a higher tax rate could mitigate air pollution and increase labor supply by encouraging households in both cities to work in city $a$, while also, on the other hand, reducing labor demand. The inverted U-shaped equilibrium labor $L_a$ and $L_{aa}$, and the U-shaped $L_{ab}$ show that the former effect dominates when the tax rate is slightly above zero and is outweighed by the latter effect as the tax rate increases further.

Since $L_{ab}$ decreases initially when the tax rate is low, $L_b$ decreases and the wage rate in city $b$ increases. The household located in city $b$ benefits from the lower labor supply in the city and is willing to supply more labor locally. Moreover, the lower emission stock in city $a$ also attracts the household in city $b$ to migrate. A lower labor supply in city $b$ also implies a lower emission level $Z_b$. As the environmental tax rate becomes high, the direction of labor flow changes: labor starts to move from city $a$ to city $b$: $L_{aa}$ is decreasing and $L_{ab}$ is increasing in $p_{Z,a}$. Similarly, the higher tax rate also prevents the household in city $b$ from migrating, resulting in an increasing $L_{bb}$ and a decreasing $L_{ba}$ against $p_{Z,a}$. In contrast to the U-shaped $Lr_a$, both the path of $L_{ba}$ and $L_{bb}$ produce a hump-shaped $Lr_b$. To conclude, when the environmental tax rate is high, households in both cities prefer to work in city $b$. Eventually, the emission level $Z_b$ rises again.

In sum, levying environmental taxation always induces a tradeoff between the benefits to output through attracting labor migration and the deterioration of output through raising firms' tax burden. Interestingly, the economy of another city would also be affected by the environmental policy. We show that when the tax rate is close to 0, the former effect dominates, while the latter effect dominates as the tax rate grows. In addition, we find that if the former effect dominates in one city, the latter effect would be dominating in another city.

In this context, the tax rate that achieves the tipping point of the curves could be regarded as the optimal tax rate for the policymaker in city $a$. Unlike the previous literature (e.g., Heutel [11], Annicchiarico and Di Dio [12]), which find that a higher environmental tax rate would definitely reduce the output level, in our two-city model, imposing a environmental taxation could simultaneously mitigate air pollutant emission and stimulate output through attracting immigrants. Note that with

the optimal tax rate, the emission level $L_b$ in city $b$ could also be reduced, but output in city $b$ will deteriorate.

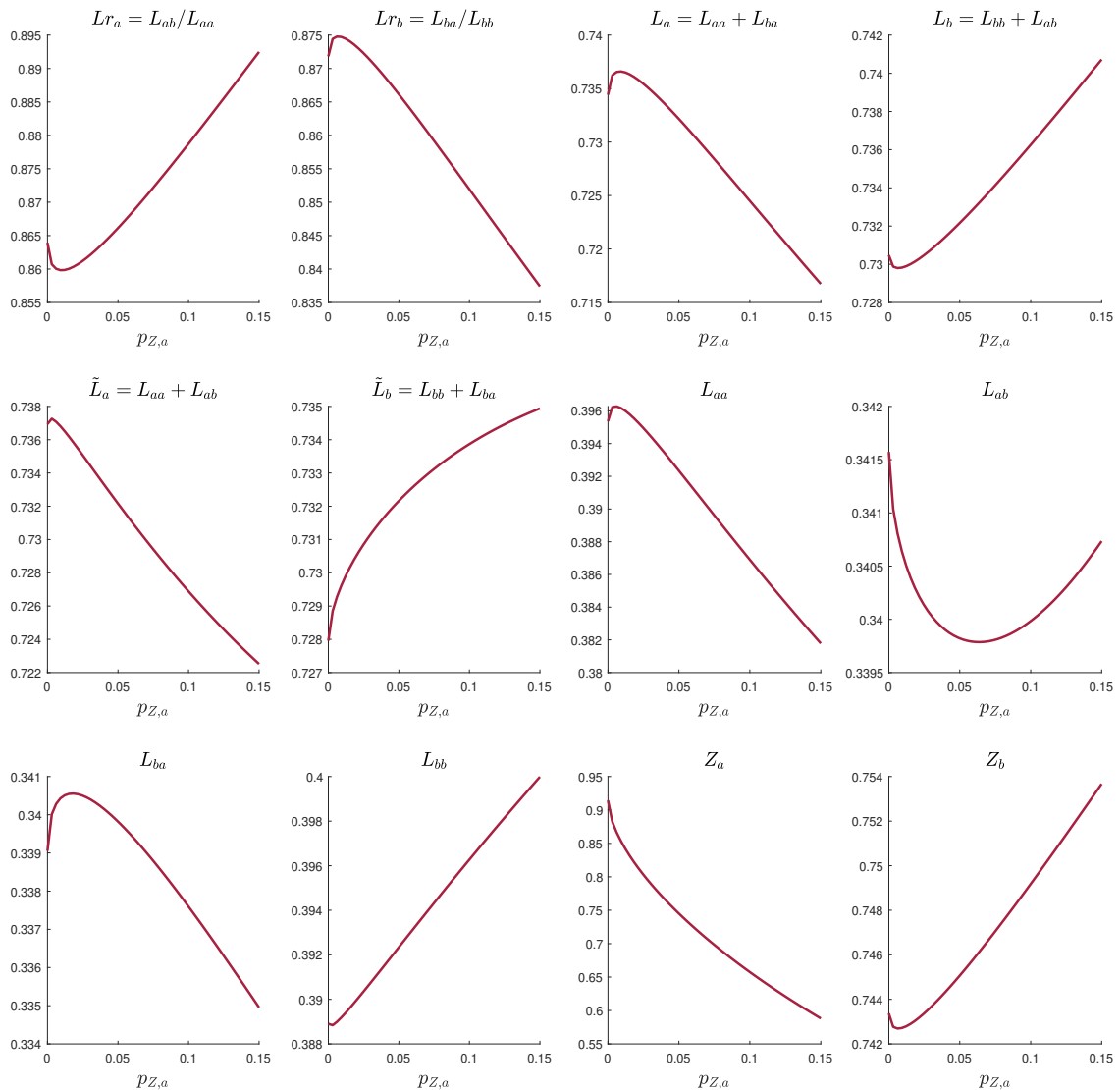

**Figure 3.** Steady-state values of labor and air pollutant emission levels in cities $a$ and $b$ against the environmental tax rate $p_{Z,a}$.

### 4.3. Short-Run Analysis

To solve for the optimal environmental tax rate that maximizes household welfare in the short run, we follow the approach of Annicchiarico and Di Dio [12] and Heutel [11] in assuming the presence of a Ramsey social planner in city $a$. The Ramsey social planner chooses a time-varying environmental tax rate and all the endogenous variables to maximize the expected lifetime utility of the representative household in city $a$, subject to all the first-order conditions in the decentralized economy stated in Section 3. The detailed formulation of the Ramsey problem can be found in the Appendix B. Unlike Annicchiarico and Di Dio [12] and Heutel [11], the economy in our model is open, and so, the Ramsey social planner in city $a$ has to respond to shocks occur in both cities. We are interested in computing the optimal environmental tax rates in the presence of these shocks, and the corresponding emission levels and labor supplies in the cities.

The four panels in Figure 4 plot the dynamics of the optimal tax rate for abatement inefficiency shocks and TFP shocks in both cities. Surprisingly, it is optimal for the policymaker to reduce the environmental tax rate to one standard deviation of positive abatement inefficiency shocks in both

cities. On the other hand, the lower panels suggest that the tax rate $p_{Z,a}$ should increase under a positive TFP shock in city $a$, while it should decrease under a positive TFP shock in city $b$. The details of the impacts induced by the tax rates under the four scenarios will be discussed below.

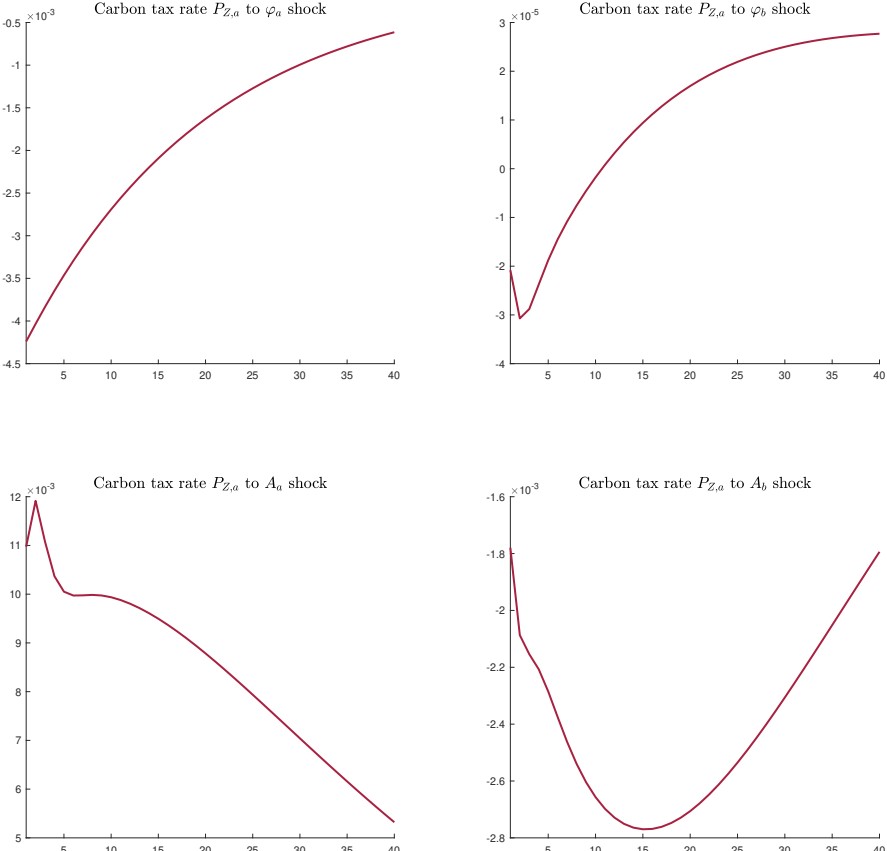

**Figure 4.** Impulse response functions (IRFs) of the optimal environmental tax rate $p_{Z,a}$ to one standard deviation of abatement inefficiency $\varphi_i$ and total factor productivity (TFP) $A_i$ shocks in city $i \in \{a, b\}$.

*4.4. Abatement Inefficiency Shocks*

Figures 5 and 6 plot the IRFs of labor and emissions levels to the abatement inefficiency shocks in cities $a$ and $b$, respectively. The solid and dashed lines are the IRFs with the optimal tax rate and with a constant tax rate $p_{Za} = 0.05$, respectively.

First, it is evident that the response of all types of labor is more stable under the optimal tax rate, compared to that under the constant tax rate. In the presence of a positive abatement inefficiency shock in city $a$, more air pollutant is emitted in the city. The higher emission level $Z_a$ and emission stock $M_a$ increase the labor disutility of working in city $a$. As a result, $L_{aa}$ and $L_{ba}$ decrease. The decrease in aggregate labor supply in city $a$ would substantially reduce output and firms' profit. This situation brings an adverse effect on household utility in city $a$. Therefore, the Ramsey social planner would find it optimal to reduce the environmental tax rate so as to stimulate output through reducing firms' tax burden. The lower environmental tax rate could increase labor demand in city $a$ and, as a result, lower the extent of the reduction of the equilibrium labor $L_a$. As shown in Figure 5, compared to the constant tax rate regime, when the optimal environmental tax rate is imposed, the household in city $a$ is less likely to migrate, and on the other hand, the household in city $b$ chooses to assign more labor in city $a$ due to the higher wage rate offered there. As shown, the lower environmental tax rate could preserve households' labor supply in both cities. In particular, total household labor supply in city $a$ $\tilde{L}_a$ decreases in a much lesser extent under the Ramsey scenario. Although the lower tax rate stimulates output in city $a$, the output of city $b$ will deteriorate. Indeed, if the policymaker in city $a$ keeps the

environmental tax rate constant, a positive abatement inefficiency shock in city $a$ would benefit the economy of city $b$ due to the higher emission stock $M_a$. However, the lower environmental tax rate $p_{Z,a}$ could counteract such an adverse effect, and attract labor from city $b$. Eventually, labor supply and output in city $b$ decrease, compared to the constant tax rate scenario. Ironically, the Ramsey problem suggests that in the optimum, the emission level in city $a$ should be higher, while the emission level in city $b$ should be lower.

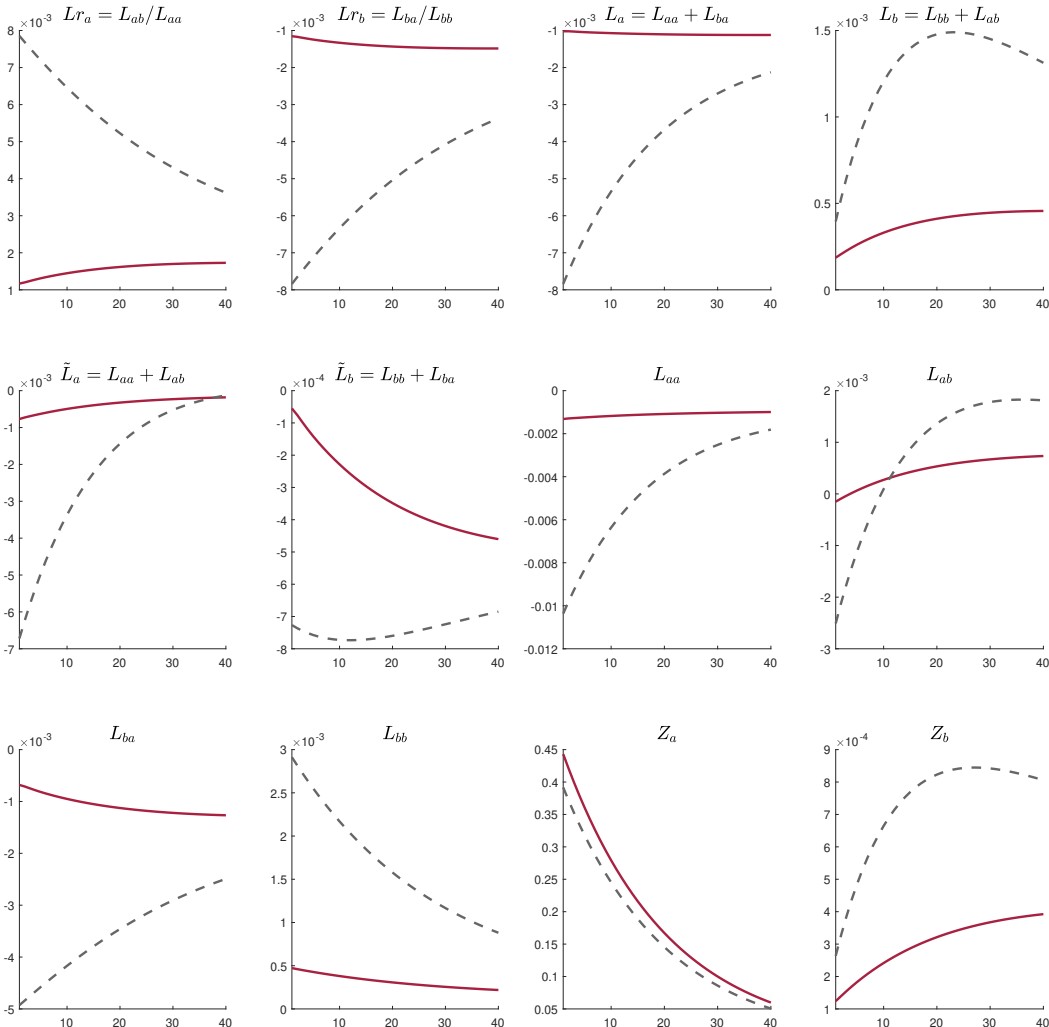

**Figure 5.** Impulse response functions (IRFs) of labor and air pollutant emissions levels in both cities to one standard deviation of abatement inefficiency shock $\varphi_a$. The solid and dashed lines are the IRFs under the optimal tax rate and a constant tax rate $p_{Z,a} = 0.05$, respectively. The values of the impulse response functions are in percent.

It is intuitive to think that the effect of $\varphi_b$ shock would bring an opposite effect on the environmental tax rate $p_{Za}$ to that of the $\varphi_a$ shock. However, the upper right panel of Figure 4 shows that the social planner still reduces the environmental tax rate in response to the $\varphi_b$ shock. Note that the $\varphi_b$ shock is unfavorable to the household in city $a$ for two reasons. First, it increases the disutility of working in city $b$, and therefore reduces household labor and income earned from city $b$. Second, a household in city $b$ chooses to assign more labor to city $a$ and compete with local workers. Output in city $a$ increases substantially after the $\varphi_b$ shock occurs, owing to the increase in the labor supply from households that switch their labor from city $b$ to city $a$. However, the local household is worse off. In this regard, the Ramsey planner of city $a$ would choose to reduce the environmental tax

rate. The lower environmental tax rate stimulates the firms in city $a$ to hire more workers, so the wage rate $w_a$ and household income in city $a$ could be maintained.

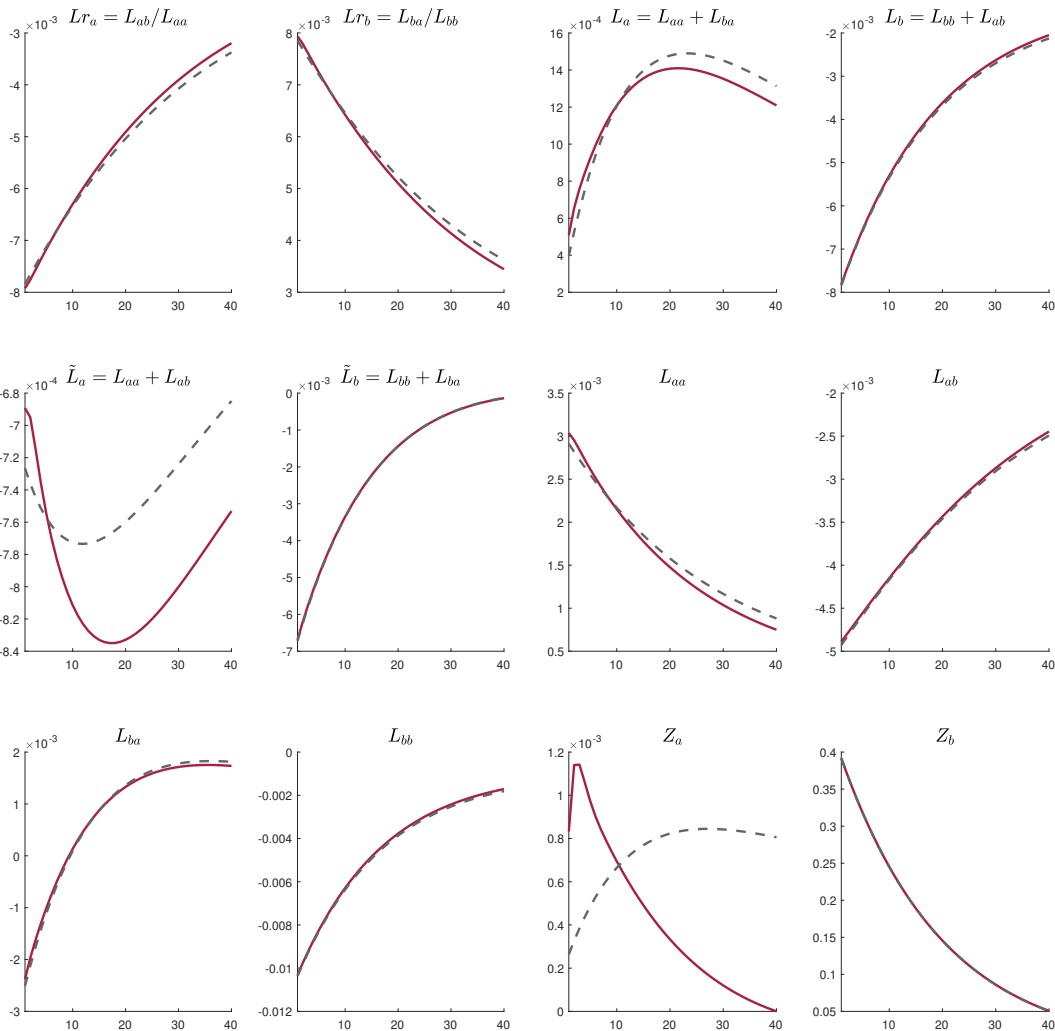

**Figure 6.** Impulse response functions (IRFs) of labor and air pollutant emissions levels in both cities to one standard deviation of abatement inefficiency shock $\varphi_b$. The solid and dashed lines are the IRFs under the optimal tax rate and a constant tax rate $p_{Z,a} = 0.05$, respectively. The values of the impulse response functions are in percent.

## 4.5. Total Factor Productivity

According to the results from the previous subsection, one might conclude that it is always optimal for the Ramsey social planner to reduce the environmental tax rate under all types of shocks. However, this is not the case. The lower left panel of Figure 4 shows that it is optimal to increase the environmental tax rate under a positive TFP shock in city $a$. Figure 7 plots the corresponding IRFs of labor and emission level in both cities to a positive TFP shock $Z_a$. Although the household in city $a$ can benefit from the positive TFP shock in its own city, the rise in the utility level is not that large in our setting, compared to that of Annicchiarico and Di Dio [12], for two reasons. First, the higher emission stock $M_a$ increases households' disutility of working in city $a$. Second, the rise in number of migrant workers from city $b$ increases the total labor supply of city $a$ and so drives down the wage rate $w_a$. As shown in the figure, total labor supplied by the household in city $b$ $\tilde{L}_b$ increases, while total labor supplied by the household in city $a$ $\tilde{L}_a$ decreases after the first few periods. This implies that the world total labor income goes dramatically to the household in city $b$. Enjoying the benefit of the TFP shock, the household in city $a$ is less willing to work due to the positive income effect. On the other hand, the household in city $b$

assigns more labor in city *a*, which would naturally reduce $L_{bb}$. Note that the inflow of labor from the household in city *b* is unfavorable to the household in city *a*. In this context, the Ramsey social planner finds it optimal to increase the environmental tax rate. Not only does a higher environmental tax rate mitigate air pollution and reduce labor disutility, but it also discourages workers who migrated from city *b*. As shown in the figure, compared with the constant tax rate regime, the responses of labor $L_{bb}$ and $L_{ba}$ are dampened under the Ramsey scenario; in contrast, labor $L_{aa}$ drops slightly and the labor $L_{ab}$ increases for most of the periods. The changes in labor supply reveal that the higher environmental tax rate is capable of reducing the surge in the labor supply from the household in city *b*. In addition, the emission level $Z_a$ decreases substantially in the presence of a higher environmental tax rate, while the extent of reduction of the emission level $Z_b$ is more limited.

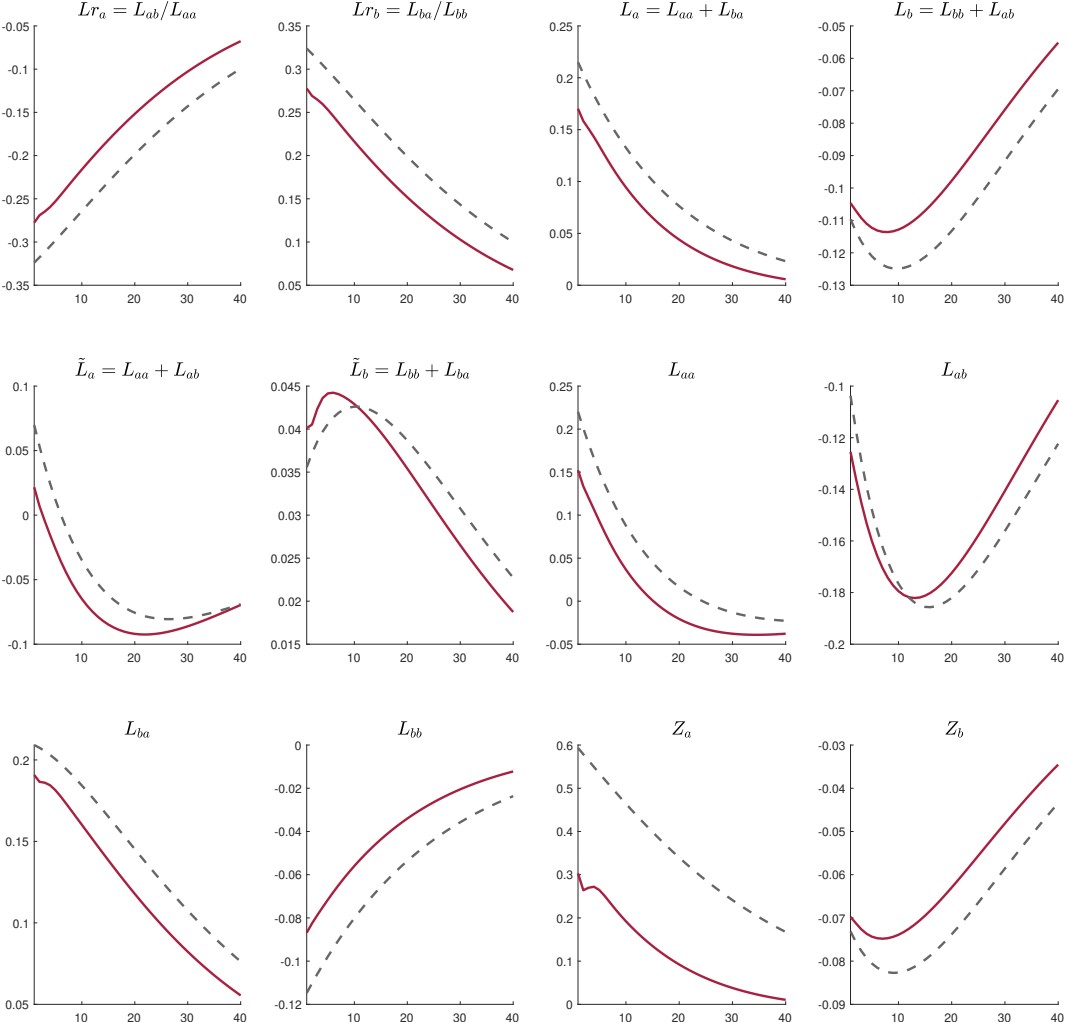

**Figure 7.** Impulse response functions (IRFs) of labor and air pollutant emissions levels in both cities to one standard deviation of TFP shock $A_a$. The solid and dashed lines are the IRFs under the optimal tax rate and a constant tax rate $p_{Z,a} = 0.05$, respectively. The values of the impulse response functions are in percent.

In our strategic setting, the Ramsey social planner has to adjust the environmental tax rate even if the shock does not occur in his/her own city. How should the Ramsey social planner respond if his opponent city enjoys a positive TFP shock? As revealed by the lower right panel of Figure 4, the environmental tax rate should be reduced in this case. This response is intuitive, since a lower environmental tax rate could stimulate firms' production and labor demand, which can therefore compete the labor with city *b*. To see this, Figure 8 plots the IRFs of emission levels and different types

of labor in both cities under a positive TFP shock $Z_b$ that occurs in city $b$. Although the IRFs of the solid and dashed lines do not diverge largely, it is still evident that the reduction in the environmental tax rate is helpful in attracting households in both cities to stay in city $a$. As a result, the magnitude of the reduction in the aggregate labor supply in city $a$ $L_a$ is smaller under the Ramsey scenario. The main adverse impact of this strategy is the change in the emission level $Z_a$. As shown, the reduction of $Z_a$ is far less than under the constant tax rate scenario.

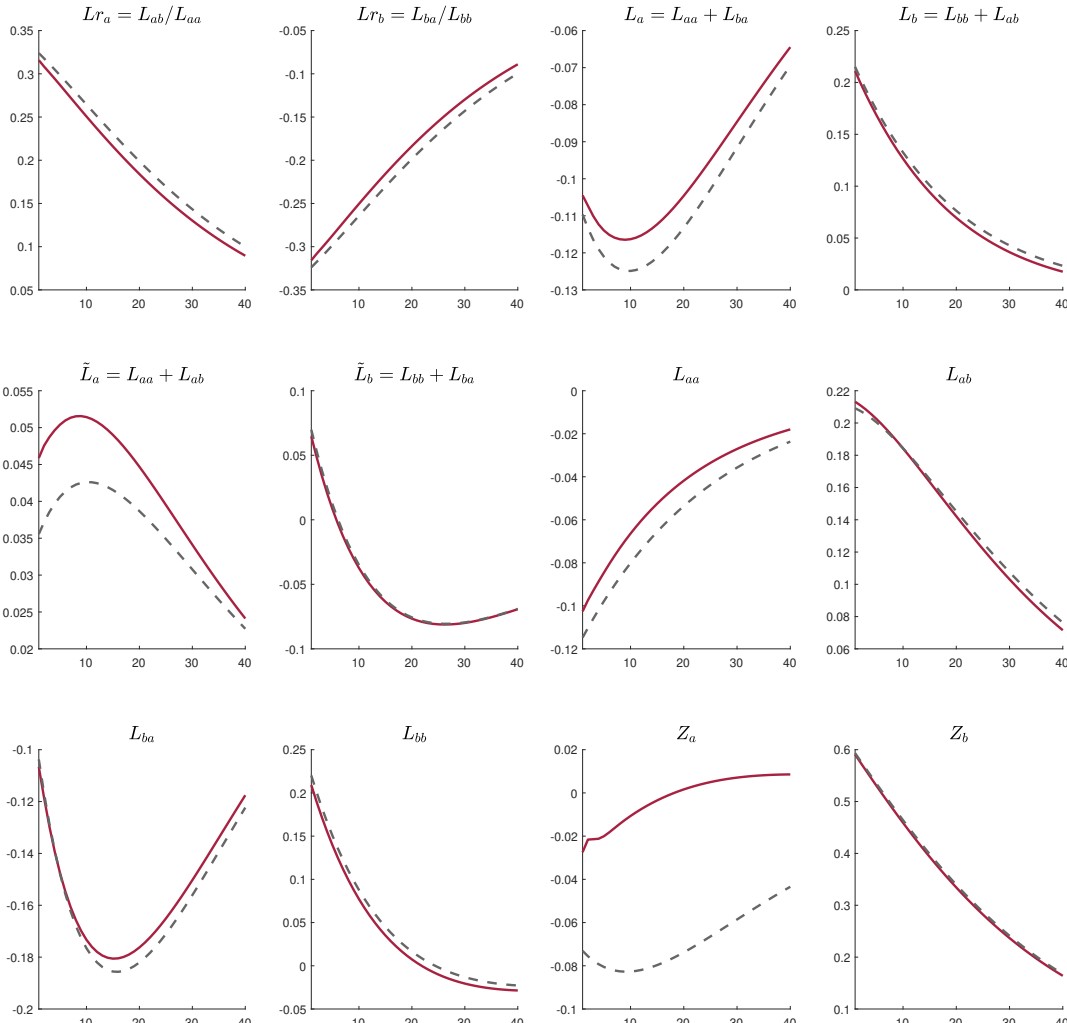

**Figure 8.** IRFs of labor and air pollutant emissions levels in both cities to one standard deviation of TFP shock $A_b$. The solid and dashed lines are the IRFs under the optimal tax rate and a constant tax rate $p_{Z,a} = 0.05$, respectively. The values of the impulse response functions are in percent.

To conclude, we find that the optimal environmental tax rate should respond positively to the TFP shock of the social planner's own city. Put differently, this result shows that the environmental tax rate should be procyclical, which is consistent with Annicchiarico and Di Dio [12] and Heutel [11]. In addition, we complement this finding by emphasizing that it is necessary to consider the response of the environmental tax rate to a TFP shock of in the opponent city to which workers in the home city frequently migrate. The optimal environmental tax rate should respond negatively to the positive TFP shock in the other city; in other words, it should be countercyclical to the business cycles of the neighbor city. It is worth noting that in reality the business cycles among cities or countries are strongly synchronized; hence the optimal environmental tax rate should also depend on the correlation of the business cycles between the two cities.

## 5. Conclusions and Extension

Since the existing literature on environmental policy analysis ignores the impact of climate policy on labor flows, this paper fills the gap by solving the optimal environmental tax rate with consideration of labor migration. On the one hand, it is shown in the literature that environmental amenity is among individuals' most important consideration with regard to migration destination. A higher environmental tax rate could mitigate the emission level, attract more migrants and, in turn, stimulate output. On the other hand, higher environmental taxation would reduce output by raising firms' operating cost and discouraging them from hiring more labor. This effect disincentizes foreign workers to migrate. These two opposite effects are carefully examined by using a two-city E-DSGE model that is an extension of the model in Annicchiarico and Di Dio [12]. As revealed by its name, our model consists of two cities. In each city, various types of stochastic shocks are presented, and households and firms that make dynamic decisions interact, so the general equilibrium is determined. A Ramsey social planner is assumed in one city to decide the optimal environmental tax rate that maximizes household welfare in the city.

The optimal environmental tax rates solved for in our model are different from those of Annicchiarico and Di Dio [12] in three ways. First, the optimal environmental tax rate should be more procyclical if the Ramsey social planner takes the effect of labor migration into account. Second, in our setting, a higher environmental tax rate could raise output through attracting foreign labor to migrate, while it would always reduce output in the setting of Annicchiarico and Di Dio [12]. Finally, our setting allows the optimal environmental tax rate to respond to the shocks occur in the opponent city, while it only responds to local shocks in Annicchiarico and Di Dio [12]. In particular, we find that the environmental tax rate should decrease in the presence of a positive TFP shock in the opponent city to prevent local work from emigrating.

The Ramsey problem in this paper assumes that only the social planner in city $a$ decides a time-varying environmental tax rate $p_{Z,a}$ that maximizes household utility in the city, while the environmental tax rate in city $b$ is assumed to be constant. In Appendix C, we study the scenario where the social planner of city $b$ also responds to the optimal $p_{Z,a}$ strategically. That is, we compute a pair of optimal environmental tax rates $\{p_{Z,a}, p_{Z,b}\}$ that satisfies a Nash equilibrium. Further, we can also compare such a pair of environmental tax rates with the one that maximizes the household utilities of both cities simultaneously. Another extension is that the household environmental concern can also be incorporated into the objective function of the social planner. In such a case, it is expected that the optimal environmental tax rate should be higher than the one suggested in this paper.

For the long-run analysis, it is interesting to examine the role of labor migration in affecting the shape of environmental Kuznets curve (EKC). Existing theoretical works on the formulating EKC treat countries independently so that no interaction between countries is modeled. With labor migration, the shape and the turning points of EKCs should be linked between countries. For a country that lies on the right of the EKC turning point, a productivity advancement simultaneously stimulates output and reduces air pollution. This would attract labor inflow, and thus should delay the time for other countries to reach their turning points. Incorporating labor migration might also change the validity of EKC hypothesis and the corresponding optimal policy suggested by the existing literature. We leave these topics to future studies.

**Funding:** This research received no external funding.

**Acknowledgments:** The author is grateful to the editors and the anonymous referees for their comments and suggestions.

**Conflicts of Interest:** The author declares no conflict of interest.

## Appendix A

The first-order conditions of the decentralized economy are summarized as follows:
Household

$$C_{i,t} = 1/\lambda_{i,t} \tag{A1}$$

$$R_{i,t}^{-1} = \beta\mathbb{E}_t \frac{1}{\Pi_{i,t+1}} \frac{\lambda_{i,t+1}}{\lambda_{i,t}} \tag{A2}$$

$$\mu_l L_{ii,t}^{\phi} = \lambda_{i,t} w_{i,t} \tag{A3}$$

$$\mu_m L_{ij,t}^{\phi} = \lambda_{i,t} w_{j,t} \tag{A4}$$

$$q_{i,t} = 1 + \gamma_I \left( \frac{I_{i,t}}{K_{i,t}} - \delta_K \right) \tag{A5}$$

$$q_{i,t} = \beta\mathbb{E}_t \frac{\lambda_{i,t+1}}{\lambda_{i,t}} \left[ r_{i,K,t+1} - \frac{\gamma_I}{2} \left( \frac{I_{i,t+1}}{K_{i,t+1}} - \delta_K \right)^2 + \gamma_I \left( \frac{I_{i,t+1}}{K_{i,t+1}} - \delta_K \right) \frac{I_{i,t+1}}{K_{i,t+1}} + (1 - \delta_K) q_{i,t+1} \right] \tag{A6}$$

$$K_{t+1} = (1 - \delta_K) K_t + I_t \tag{A7}$$

Firms

$$L_{i,t} = L_{ai,t} + L_{bi,t} \tag{A8}$$

$$Y_{i,t} = (1 - \Upsilon(M_{i,t})) A_{i,i} K_{i,t}^{\alpha} L_{i,t}^{1-\alpha} (D_{i,p,t})^{-1} \tag{A9}$$

$$(1 - \alpha)(1 - \Upsilon(M_{i,t})) A_{i,t} K_{i,t}^{\alpha} L_{i,t}^{-\alpha_N} \Psi_{i,t} = w_{i,t} \tag{A10}$$

$$\alpha(1 - \Upsilon(M_{i,t})) A_{i,t} K_{i,t}^{\alpha-1} L_{i,t}^{1-\alpha} \Psi_{i,t} = r_{i,K,t} \tag{A11}$$

$$\varphi p_{Z,i,t} = \phi_1 \phi_2 U_{i,t}^{\phi_2-1}$$

Calvo pricing

$$p_{i,t}^* = \frac{\theta}{\theta-1} \frac{X_{i,t} + \Omega_{i,t}}{\Theta_{i,t}} \tag{A12}$$

$$X_{i,t} = \lambda_{i,t} \Psi_{i,t} Y_{i,t} + \nu\beta\mathbb{E}_t 1 X_{i,t+1} \tag{A13}$$

$$\Theta_{i,t} = \lambda_{i,t} Y_{i,t} + \nu\beta\mathbb{E}_t \Pi_{i,t+1}^{\theta-1} \Theta_{i,t+1} \tag{A14}$$

$$\Omega_{i,t} = \lambda_{i,t} \left[ \phi_1 U_{i,t}^{\phi_2} + p_{Z,i,t}(1 - U_{i,t})\varphi \right] Y_{i,t} + \xi\beta\mathbb{E}_t \Pi_{i,t+1}^{\theta} \Omega_{i,t+1} \tag{A15}$$

$$1 = \nu\Pi_{i,t}^{\theta-1} + (1 - \xi)(p_{i,t}^*)^{1-\theta} \tag{A16}$$

$$D_{i,p,t} = (1 - \nu) p_{i,t}^{*-\theta} + \nu\Pi_{i,t}^{\theta} D_{i,p,t-1} \tag{A17}$$

$$MC_{i,t} = \Psi_{i,t} + \phi_1 U_{i,t}^{\phi_2} + p_{i,Z,t}(1 - U_{i,t})\varphi \tag{A18}$$

Carbon emissions and abatement cost

$$Z_{i,t} = \varphi_{i,t}(1 - U_{i,t}) Y_{i,t} D_{i,p,t} \tag{A19}$$

$$M_{i,t} = (1 - \delta_M) M_{i,t-1} + Z_{i,t} + Z^* \tag{A20}$$

$$C_{A,i,t} = \phi_1 U_{i,t}^{\phi_2} Y_{i,t} D_{i,p,t} \tag{A21}$$

Public sector and nominal interest rate

$$T_{i,t} + p_{i,Z,t} Z_{i,t} = G_i \tag{A22}$$

$$R_t = R \left(\frac{\Pi_t}{\Pi}\right)^{\iota_\pi} \left(\frac{Y_t}{Y}\right)^{\iota_Y} \tag{A23}$$

Good market equilibrium

$$Y_{i,t} - w_{i,t}L_{ji,t} + w_{j,t}L_{ij,t} = C_{i,t} + I_{i,t} + G_{i,t} + \phi_1 U_{i,t}^{\phi_2} Y_{i,t} D_{i,p,t} + \frac{\gamma_I}{2}\left(\frac{I_{i,t}}{K_{i,t}} - \delta_K\right)^2 K_{i,t} \tag{A24}$$

Shocks

$$\ln A_{i,t} = (1 - \rho_A)\ln A + \rho_A \ln A_{i,t-1} + \varepsilon_{A,i,t} \tag{A25}$$

$$\log(\varphi_{i,t}/\varphi) = \rho_{\varphi_i}\log(\varphi_{i,t-1}/\varphi) + \varepsilon_{\varphi_i,t} \tag{A26}$$

## Appendix B. Ramsey Problem

Assume that there is a social planner who chooses the path of carbon tax rate $p_{Za,t}$ and other endogenous variables, so as to maximize expected household lifetime utility, subject to the equilibrium conditions of the decentralized economy shown in Appendix A. In particular, the Lagrangian function is

$$\mathcal{L}_t = \sum_{t=0}^{\infty}\beta^t\left(\ln C_{a,t} - \mu_{l,a}\frac{L_{aa,t}^{1+\phi}}{1+\phi} - \mu_{m,a}\frac{L_{ab,t}^{1+\phi}}{1+\phi}\right)$$

$$+\lambda_{1,t}(\lambda_{a,t} - 1/C_{a,t})$$

$$+\lambda_{2,t}\left[R_{a,t}^{-1} - \beta\mathbb{E}_t\frac{1}{\Pi_{a,t+1}}\frac{\lambda_{a,t+1}}{\lambda_{a,t}}\right]$$

$$+\lambda_{3,t}(\mu_{l,a}L_{aa,t}^\phi - \lambda_{a,t}w_{a,t})$$

$$+\lambda_{4,t}(\mu_{m,a}L_{ab,t}^\phi - \lambda_{a,t}w_{b,t})$$

$$+\lambda_{5,t}\left\{\lambda_{a,t}q_{a,t} - \beta\mathbb{E}_t\lambda_{a,t+1}\left[r_{a,K,t+1} - \frac{\gamma_I}{2}\left(\frac{I_{a,t+1}}{K_{a,t+1}} - \delta_K\right)^2 + \gamma_I\left(\frac{I_{a,t+1}}{K_{a,t+1}} - \delta_K\right)\frac{I_{a,t+1}}{K_{a,t+1}} + (1-\delta_K)q_{a,t+1}\right]\right\}$$

$$+\lambda_{6,t}\left[q_{a,t} - 1 - \gamma_I\left(\frac{I_{a,t}}{K_{a,t}} - \delta_K\right)\right]$$

$$+\lambda_{7,t}\left[Y_{a,t} - w_{a,t}L_{ba,t} + w_{b,t}L_{ab,t} - \left(C_{a,t} + I_{a,t} + G_a + \phi_1 U_{a,t}^{\phi_2}Y_{a,t}D_{a,p,t} + \frac{\gamma_I}{2}\left(\frac{I_{a,t}}{K_{a,t}} - \delta_K\right)^2 K_{a,t}\right)\right]$$

$$+\lambda_{8,t}[K_{a,t+1} - ((1-\delta_K)K_{a,t} + I_{a,t})]$$

$$+\lambda_{9,t}(Y_{a,t} - (1 - Y(M_{a,t}))A_{a,t}K_{a,t}^\alpha L_{a,t}^{1-\alpha}(D_{a,p,t})^{-1})$$

$$+\lambda_{10,t}((1-\alpha)(1 - Y(M_{a,t}))A_{a,t}K_{a,t}^\alpha L_{a,t}^{-\alpha}\Psi_{a,t} - w_{a,t})$$

$$+\lambda_{11,t}(\alpha(1 - Y(M_{a,t}))A_{a,t}K_{a,t}^{\alpha-1}L_{a,t}^{1-\alpha}\Psi_{a,t} - r_{a,K,t})$$

$$+\lambda_{12,t}(\varphi p_{Z,a,t} - \phi_1\phi_2 U_{a,t}^{\phi_2-1})$$

$$+\lambda_{13,t}\left[p_{a,t}^* - \frac{\theta}{\theta-1}\frac{X_{a,t}+\Omega_{a,t}}{\Theta_{a,t}}\right]$$

$$+\lambda_{14,t}\left[X_{a,t} - \left(\frac{1}{C_{a,t}}\Psi_{a,t}Y_{a,t} + \nu\beta\mathbb{E}_t\Pi_{t+1}^\theta X_{t+1}\right)\right]$$

$$+\lambda_{15,t}\left[\Theta_{a,t} - \left(\frac{1}{C_{a,t}}Y_{a,t} + \nu\beta\mathbb{E}_t\Pi_{a,t+1}^{\theta-1}\Theta_{a,t+1}\right)\right]$$

$$+\lambda_{16,t}\left\{\Omega_{a,t} - \frac{1}{C_{a,t}}\left[\phi_1 U_{a,t}^{\phi_2} + p_{Z,a,t}(1-U_{a,t})\varphi\right]Y_{a,t} - \nu\beta\mathbb{E}_t\Pi_{a,t+1}^\theta\Omega_{a,t+1}\right\}$$

$$+\lambda_{17,t}\left[1 - (\nu\Pi_{a,t}^{\theta-1} + (1-\nu)(p_{a,t}^*)^{1-\theta})\right]$$

$$+\lambda_{18,t}(D_{a,p,t} - (1-\nu)p_{a,t}^{*-\theta} - \nu\Pi_{a,t}^\theta D_{a,p,t-1})$$

$$+\lambda_{19,t}(Z_{a,t} - (1-U_{a,t})\varphi Y_{a,t}D_{a,p,t})$$

$$+\lambda_{20,t}((M_{a,t} - (1-\delta_M)M_{a,t-1} - Z_{a,t} - Z^*)$$

$$+\lambda_{21,t}(R_{a,t} - R\left(\frac{\Pi_{a,t}}{\Pi}\right)^{\iota_\pi}\left(\frac{Y_{a,t}}{Y}\right)^{\iota_Y})$$

$$+\lambda_{22,t}(L_{a,t} - L_{aa,t} - L_{ba,t})$$

The choice variables are $\{C_{a,t}, D_{a,p,t}, I_{a,t}, K_{a,t+1}, L_{aa,t}, L_{ab,t}, L_{a,t}, M_{a,t}, p_{a,t}^*, R_{a,t}, r_{a,K,t}, U_{a,t}, w_{a,t}, X_{a,t}, Y_{a,t}, Z_{a,t}, \Theta_{a,t}, \Pi_{a,t}, \Psi_{a,t}, \Omega_{a,t}, q_{a,t}, \lambda_{a,t}, p_{Za,t}\}$.

Below are the first-order conditions:

The first-order condition for $C_{a,t}$

$$\frac{1}{C_{a,t}} + \frac{\lambda_{1,t}}{C_{a,t}^2} - \lambda_{7,t} + \lambda_{15,a,t}\frac{Y_{a,t}}{C_{a,t}^2} + \frac{\lambda_{16,t}Y_{a,t}}{C_{a,t}^2}(\phi_1 U_{a,t}^{\phi_2} - p_{Za,t}\varphi(U_{a,t}-1)) + \lambda_{14,t}\frac{\Psi_{a,t}Y_{a,t}}{C_{a,t}^2} = 0.$$

The first-order condition for $D_{a,p,t}$

$$\lambda_{18,t} - \lambda_{18,t+1}\beta\nu\Pi_{a,t+1}^{\theta} - \lambda_{19,t}\varphi Y_{a,t}(1-U_{a,t}) - \lambda_{7,t}\phi_1 U_{a,t}^{\phi_2}Y_{a,t} + \lambda_{9,t}(1-Y(M_{a,t}))A_{a,t}K_{a,t}^{\alpha}L_{a,t}^{1-\alpha}D_{a,p,t}^{-2} = 0.$$

The first-order condition for $I_{a,t}$

$$-\lambda_{5,t-1}\lambda_t\gamma_I\frac{I_{a,t}}{K_{a,t}^2} + \lambda_{7,t}\left[\gamma_I\left(\delta_K - \frac{I_{a,t}}{K_{a,t}}\right) - 1\right] - \lambda_{8,t} - \lambda_{6,t}\frac{\gamma_I}{K_{a,t}} = 0.$$

The first-order condition for $K_{a,t+1}$

$$\begin{aligned}
&+\lambda_{5,t}\gamma_I\frac{I_{a,t+1}}{K_{a,t+1}^2}\left[\left(\frac{I_{a,t+1}}{K_{a,t+1}} - \delta_K\right)^2 - \frac{2I_{a,t+1}}{K_{a,t+1}} + \delta_K\right] + \lambda_{6,t+1}\beta\gamma_I\frac{I_{a,t+1}}{K_{a,t+1}^2} \\
&+\lambda_{7,t+1}\beta(-\frac{\gamma_I}{2}(\frac{I_{a,t+1}}{K_{a,t+1}} - \delta_K)^2 + \gamma_I(\frac{I_{a,t+1}}{K_{a,t+1}} - \delta_K)\frac{I_{a,t+1}}{K_{a,t+1}}) - \lambda_{8,t+1}\beta(1-\delta_K) + \lambda_{11,t} \\
&-\lambda_{9,t+1}\beta\alpha\frac{Y_{a,t+1}}{K_{a,t+1}} + \lambda_{10,t+1}\beta\alpha(1-\alpha)(1-Y(M_{a,t+1}))A_{a,t+1}K_{a,t+1}^{\alpha-1}L_{a,t+1}^{-\alpha}\Psi_{a,t+1} \\
&+\lambda_{11,t+1}\beta\alpha(\alpha-1)(1-Y(M_{a,t+1}))A_{a,t+1}K_{a,t+1}^{\alpha-2}L_{a,t+1}^{1-\alpha}\Psi_{a,t} = 0
\end{aligned}$$

The first-order condition for $L_{aa,t}$

$$-\mu_{l,a}L_{aa,t}^{\phi} + \lambda_{3,t}\mu_{l,a}\phi L_{aa,t}^{\phi-1} - \lambda_{22,t} = 0.$$

The first-order condition for $L_{ab,t}$

$$-\mu_{m,a}L_{ab,t}^{\phi} + \lambda_{4,t}\mu_{m,a}\phi L_{ab,t}^{\phi-1} + \lambda_{7,t}w_{b,t} = 0.$$

The first-order condition for $L_{a,t}$

$$\begin{aligned}
-\lambda_{9,t}(1-\alpha)(1-Y(M_{a,t}))A_{a,t}K_{a,t}^{\alpha}L_{a,t}^{-\alpha}(D_{a,p,t})^{-1} - \lambda_{10,t}\alpha(1-\alpha)(1-Y(M_{a,t}))A_{a,t}K_{a,t}^{\alpha}L_{a,t}^{-(1+\alpha)}\Psi_{a,t} \\
+\lambda_{11,t}\alpha(1-\alpha)(1-Y(M_{a,t}))A_{a,t}K_{a,t}^{\alpha-1}L_{a,t}^{-\alpha}\Psi_{a,t} + \lambda_{22,t} = 0
\end{aligned}$$

The first-order condition for $M_{a,t}$

$$\begin{aligned}
\lambda_{20,t} - \lambda_{20,t+1}\beta(1-\delta_M) + \lambda_{9,t}A_tK_{a,t}^{\alpha}L_{a,t}^{1-\alpha}Y'(M_{a,t})D_{a,p,t}^{-1} + \lambda_{10,t}(\alpha-1)\Psi_{a,t}A_{a,t}K_{a,t}^{\alpha}L_{a,t}^{-\alpha}Y'(M_{a,t}) \\
-\lambda_{11,t}\alpha Y'(M_{a,t})\Psi_{a,t}A_{a,t}K_{a,t}^{\alpha-1}L_{a,t}^{1-\alpha} + \lambda_{3,t}\mu_2 L_{aa,t}^{\phi} + \lambda_{4,t}\mu_2 L_{ab,t}^{\phi} = 0
\end{aligned}$$

where $Y'(M_{a,t}) = \gamma_1 + 2\gamma_2 M_{a,t}$.

The first-order condition for $p_{a,t}^*$

$$\lambda_{13,t} - \lambda_{17,t}(1-\nu)(1-\theta)p_{a,t}^{*-\theta} + \lambda_{18,t}\theta(1-\nu)p_{a,t}^{*-(1+\theta)} = 0.$$

The first-order condition for $R_{a,t}$

$$\lambda_{21,t} - \lambda_{2,t}R_{a,t}^{-2} = 0.$$

The first-order condition for $r_{a,K,t}$

$$-\lambda_{5,t-1}\lambda_{a,t} - \lambda_{11,t} = 0.$$

The first-order condition for $U_{a,t}$

$$-\lambda_{7,t}\phi_1\phi_2 U_{a,t}^{\phi_2-1}Y_{a,t}D_{a,p,t} - \lambda_{12,t}\phi_1\phi_2(\phi_2-1)U_{a,t}^{\phi_2-2}$$
$$+\lambda_{16,t}\frac{Y_{a,t}}{C_{a,t}}(p_{Za,t}\varphi - \phi_1\phi_2 U_{a,t}^{\phi_2-1}) + \lambda_{19,t}\varphi Y_{a,t}D_{a,p,t} = 0$$

The first-order condition for $w_{a,t}$

$$-\lambda_{10,t} - \lambda_{3,t}\lambda_{a,t} - \lambda_{4,t}\lambda_{a,t} - \lambda_{7,t}L_{ba,t} = 0.$$

The first-order condition for $X_{a,t}$

$$-\lambda_{13,t}\frac{\theta}{\theta-1}\frac{1}{\Theta_{a,t}} + \lambda_{14,t} - \lambda_{14,t-1}\nu\Pi_{a,t}^{\theta} = 0.$$

The first-order condition for $Y_{a,t}$

$$-\lambda_{7,t}(\phi_1 D_{a,p,t}U_{a,t}^{\phi_2} - 1) + \lambda_{9,t} - \lambda_{14,t}\frac{\Psi_{a,t}}{C_{a,t}} - \lambda_{15,t}\frac{1}{C_{a,t}}$$
$$-\lambda_{16,t}\frac{1}{C_{a,t}}(\phi_1 U_{a,t}^{\phi_2} - \varphi p_{Za,t}(U_{a,t}-1)) - \lambda_{19,t}\varphi D_{a,p,t}(1-U_{a,t}) - \lambda_{21,t}\iota_Y\frac{R_{a,t}}{Y_{a,t}} = 0$$

The first-order condition for $Z_{a,t}$

$$\lambda_{19,t} = \lambda_{20,t}.$$

The first-order condition for $\Theta_{a,t}$

$$\lambda_{15,t} - \lambda_{15,t-1}\nu\Pi_{a,t}^{\theta-1} + \lambda_{13,t}\frac{\theta}{\theta-1}\frac{\Omega_{a,t} + X_{a,t}}{\Theta_{a,t}^2} = 0.$$

The first-order condition for $\Pi_{a,t}$

$$\lambda_{2,t-1}\frac{\lambda_{a,t}}{\lambda_{a,t-1}\Pi_{a,t}^2} + \lambda_{14,t-1}(-\nu\theta\Pi_{a,t}^{\theta-1}X_{a,t}) + \lambda_{15,t-1}(-(\theta-1)\nu\Pi_{a,t}^{\theta-2}\Theta_{a,t})$$
$$+\lambda_{16,t-1}(-\nu\theta\Pi_{a,t}^{\theta-1}\Omega_{a,t}) + \lambda_{17,t}(-\nu(\theta-1)\Pi_{a,t}^{\theta-2}) + \lambda_{18,t}(-\nu\theta\Pi_{a,t}^{\theta-1}D_{a,p,t-1})$$
$$-\lambda_{21,t}\iota_\pi\frac{R_{a,t}}{\Pi_{a,t}} = 0$$

The first-order condition for $\Psi_{a,t}$

$$\lambda_{10,t}(1-\alpha)(1-\Upsilon(M_{a,t}))A_{a,t}K_{a,t}^\alpha L_{a,t}^{-\alpha} + \lambda_{11,t}\alpha(1-\Upsilon(M_{a,t}))A_{a,t}K_{a,t}^{\alpha-1}L_{a,t}^{1-\alpha} - \lambda_{14,t}\frac{Y_{a,t}}{C_{a,t}} = 0.$$

The first-order condition for $\Omega_{a,t}$

$$-\lambda_{13,t}\frac{\theta}{\theta-1}\frac{1}{\Theta_{a,t}} + \lambda_{16,t} - \lambda_{16,t-1}\nu\Pi_{a,t}^{\theta} = 0.$$

The first-order condition for $q_{a,t}$

$$\lambda_{5,t}\lambda_t - \lambda_{5,t-1}\lambda_t(1-\delta_K) + \lambda_{6,t} = 0.$$

The first-order condition for $\lambda_{a,t}$

$$\lambda_{1,t} - \lambda_{2,t-1}\frac{1}{\Pi_{a,t}\lambda_{t-1}\beta} + \lambda_{2,t}\frac{\beta\lambda_{t+1}}{\Pi_{a,t+1}\lambda_t^2} - \lambda_{3,t}w_{a,t} - \lambda_{4,t}w_{b,t}$$
$$+\lambda_{5,t}q_{a,t} - \lambda_{5,t-1}\left[r_{a,K,t} - \frac{\gamma_I}{2}\left(\frac{I_{a,t}}{K_{a,t}} - \delta_K\right)^2 + \gamma_I\left(\frac{I_{a,t}}{K_{a,t}} - \delta_K\right)\frac{I_{a,t}}{K_{a,t}} + (1-\delta_K)q_{a,t}\right] = 0$$

The first-order condition for $p_{Z,a,t}$

$$\lambda_{12,t}\varphi - \lambda_{16,t}\frac{Y_{a,t}}{C_{a,t}}(1 - U_{a,t})\varphi = 0.$$

**Appendix C. Robustness Check and Model Extensions**

In this Appendix, to ensure the robustness of the result, we follow Annicchiarico and Di Dio [12] to vary the value of the Calvo pricing parameter $\nu$ to see how do the optimal environmental rates depend on the value of $\nu$.

Then, we extend our model into two ways. First, we compare the IRFs of the macroeconomic variables under the optimal tax rates to those under a cap-and-trade regime, instead of to those under a constant environmental tax rate as shown in Figures 5–8. Second, we assume there are Ramsey social planners in both cities, and solve for the non-cooperative optimal environmental tax rates under Nash equilibrium.

*Appendix C.1. Optimal Environmental Tax Rate Under Different Degree of Price Stickiness*

As emphasized by Annicchiarico and Di Dio [12], the degree of the price stickiness plays an important role in determining the optimal environmental policy. In this regard, the short-run analysis in Section 4.3 is re-examined in this section with different values of $\nu$ (It is noting that the long-run analysis in Section 4.2 is not affected by the value of $\nu$, since in the deterministic steady state, the intermediate good price would become $p = \theta/(\theta-1)$ for any degree of price stickiness $\nu$.). Recall that the degree of the price stickiness in our model is solely controlled by the Calvo pricing parameter $\nu$. Since in any period, there is only $1-\nu$ portion of firms can re-adjust their price levels, a lower value of $\nu$ implies a higher price flexibility.

Figures A1–A5 show the optimal environmental tax rates and the corresponding IRFs of labor under the four shocks considered in the model with different values of $\nu$. In particular, in each figure, the solid, dashed, and dotted lines represent the paths when $\nu$ equals 0.75, 0.5, and 0.85, respectively. Figure A1 shows the trajectories of the optimal environmental tax rates under different values of $\nu$. In general, a higher degree of price stickiness raises the volatility of the IRFs of the optimal environmental tax rates. Taking the IRFs to the $\varphi_a$ shock as an example. As discussed in Section 4.3, since a positive $\varphi_a$ shock increases air pollutant emissions and discourages employment in the city, the Ramsey social planner finds it optimal to reduce the environmental tax rate so as to counteract such a negative effect. A lower environmental tax rate reduces the firms' operating cost and thus can incentivize the firms to hire more workers. If the price is flexible, the good price level would decrease in response to the $\varphi_a$ shock. This price movement could stimulate the good demand and insulate part of the negative impacts of the shock. Hence, the extent of the environmental tax rate to which the Ramsey social planner need to manipulate is reduced.

It is worth noting that the optimal environmental tax rate even increases to the $\varphi_b$ shock when $\nu = 0.85$. Further, the optimal environmental tax rate increases to the TFP shock in city $b$ when $\nu = 0.5$. The IRFs of labor and air pollutant emissions shown from Figures A2–A5 with different values of $\nu$ move in the same direction, but vary with different magnitude.

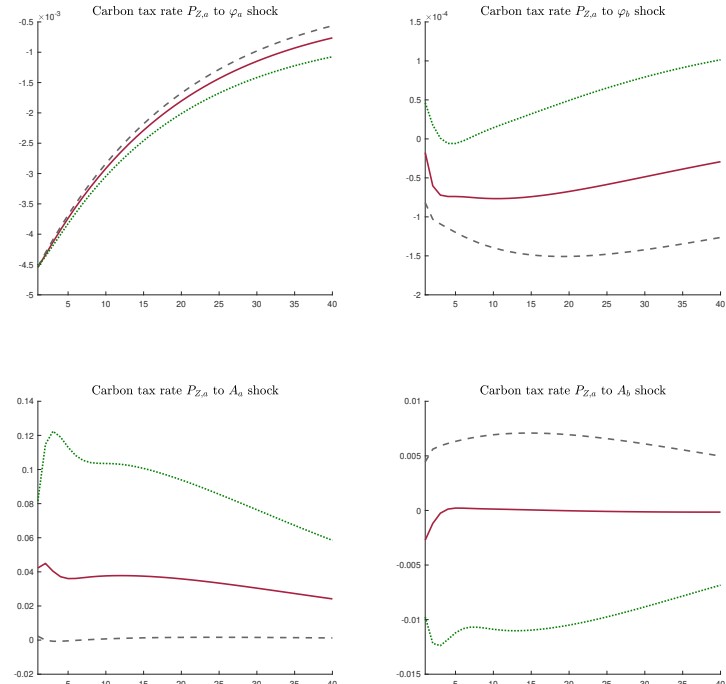

**Figure A1.** IRFs of the optimal environmental tax rate $p_{Z,a}$ to one standard deviation of abatement inefficiency $\varphi_i$ and TFP $A_i$ shocks in city $i \in \{a, b\}$. The solid, dashed, and dotted lines are the IRFs under the optimal tax rate when $\eta = 0.75$, 0.5 and 0.85, respectively.

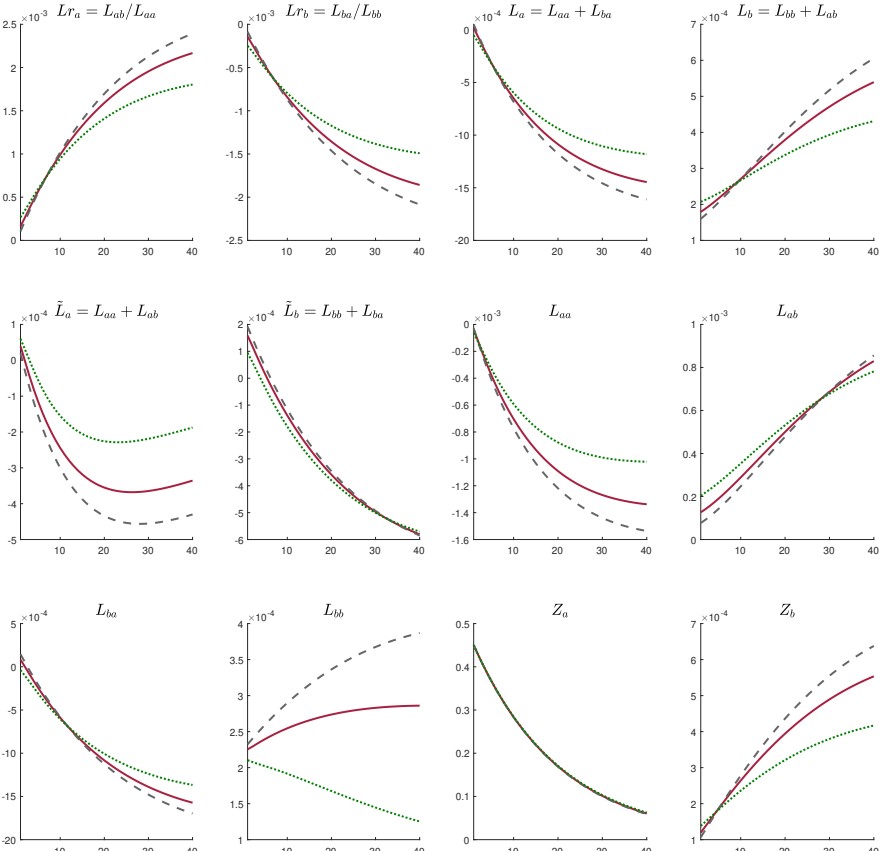

**Figure A2.** IRFs of labor and air pollutant emissions levels in both cities to one standard deviation of abatement inefficiency shock $\varphi_a$. The solid, dashed, and dotted lines are the IRFs under the optimal tax rate when $\eta = 0.75$, 0.5 and 0.85, respectively. The values of the impulse response functions are in percent.

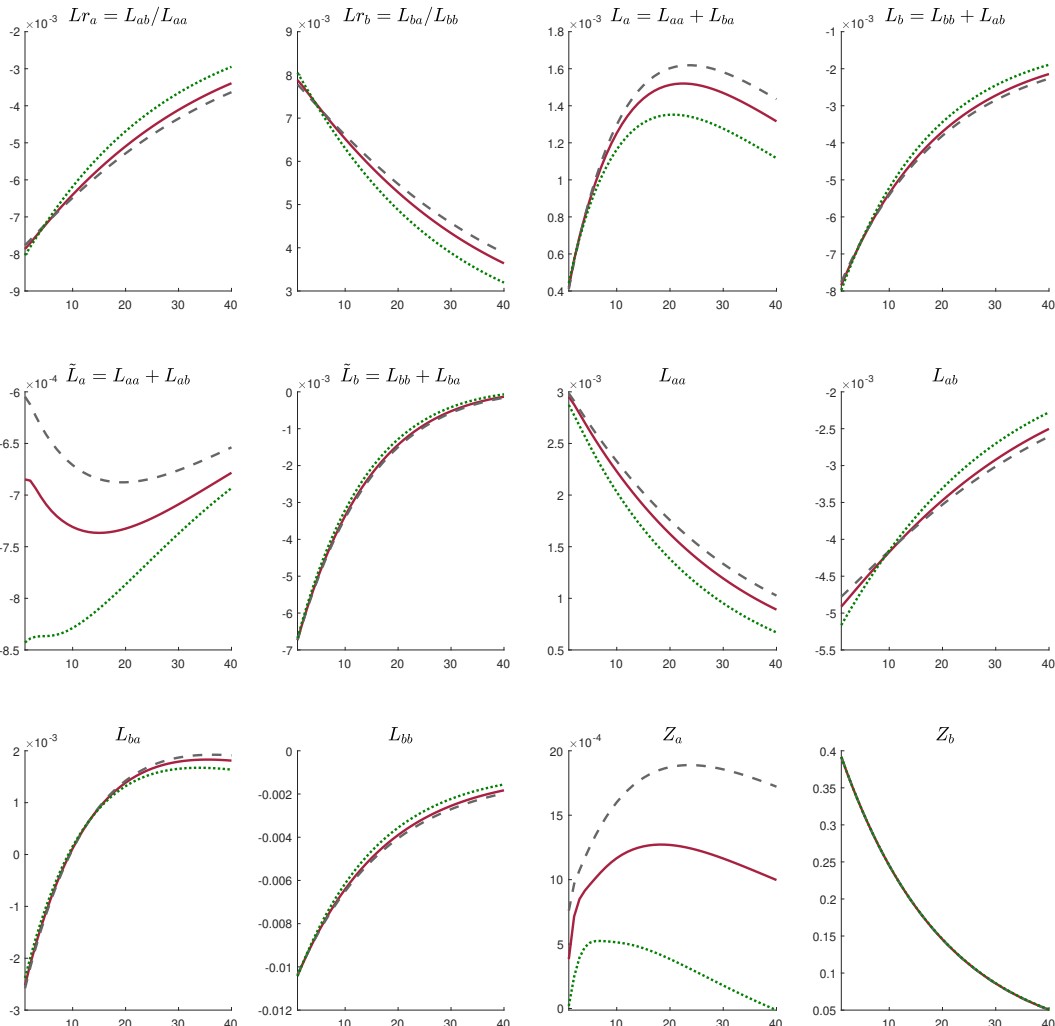

**Figure A3.** IRFs of labor and air pollutant emissions levels in both cities to one standard deviation of abatement inefficiency shock $\varphi_b$. The solid, dashed, and dotted lines are the IRFs under the optimal tax rate when $\eta = 0.75$, 0.5 and 0.85, respectively. The values of the impulse response functions are in percent.

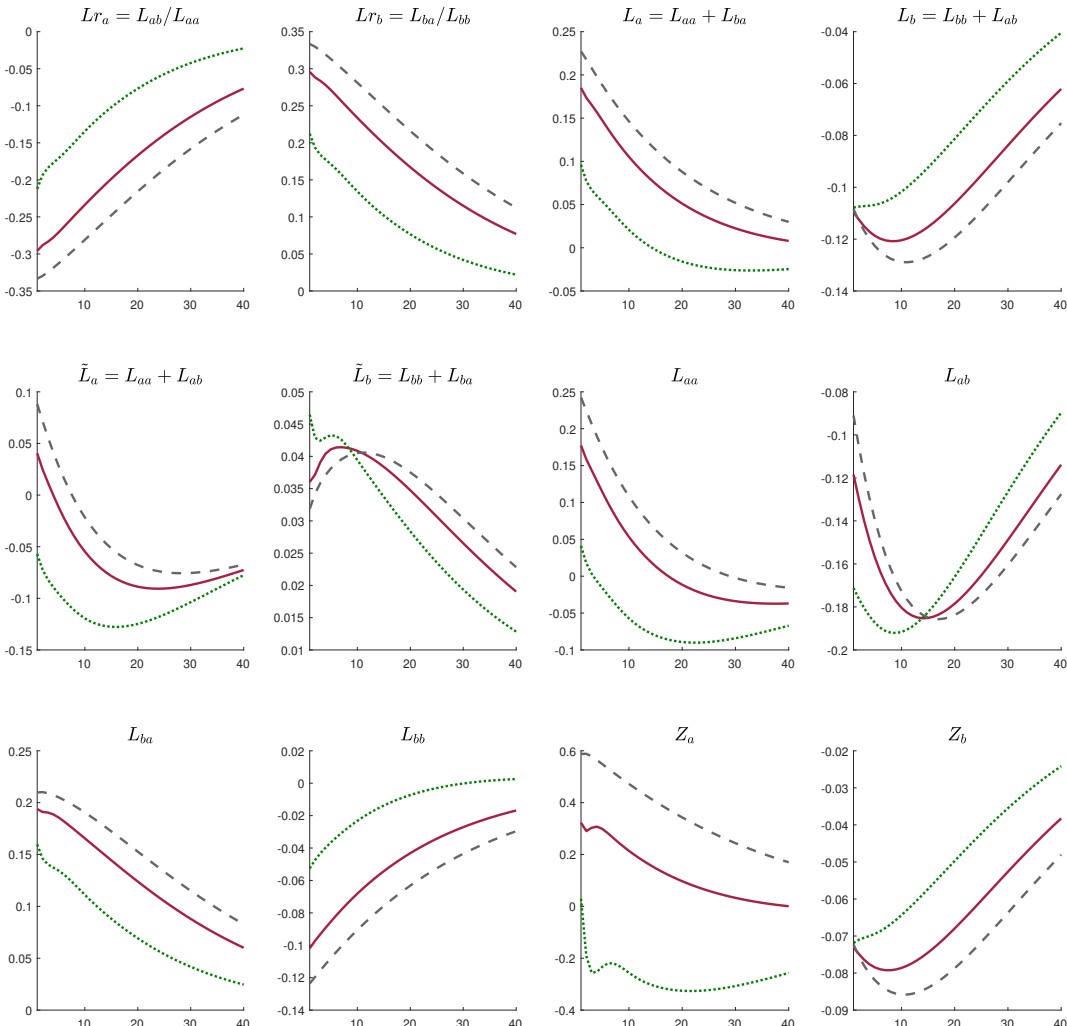

**Figure A4.** IRFs of labor and air pollutant emissions levels in both cities to one standard deviation of TFP shock $A_a$. The solid, dashed, and dotted lines are the IRFs under the optimal tax rate when $\eta = 0.75, 0.5$ and $0.85$, respectively. The values of the impulse response functions are in percent.

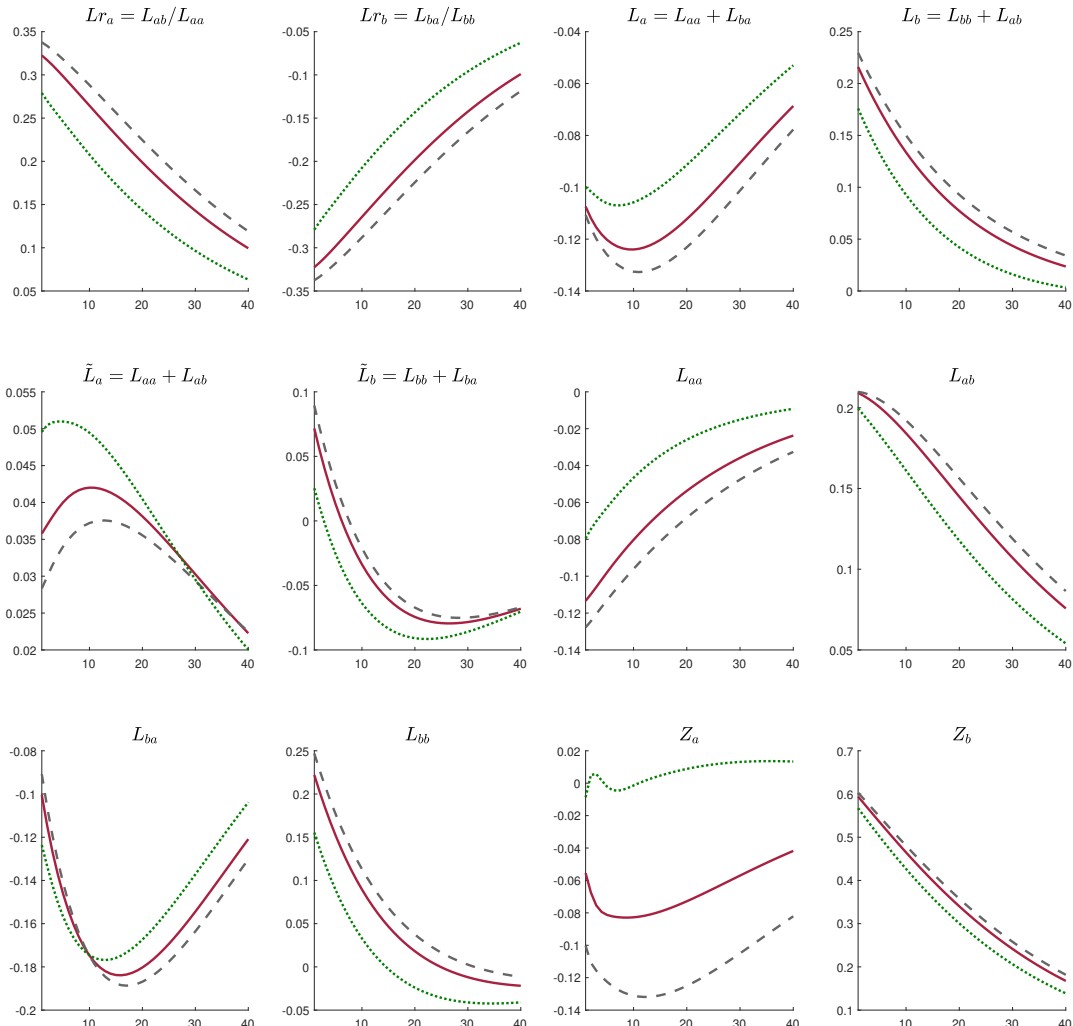

**Figure A5.** IRFs of labor and air pollutant emissions levels in both cities to one standard deviation of TFP shock $A_b$. The solid, dashed, and dotted lines are the IRFs under the optimal tax rate when $\eta = 0.75$, 0.5 and 0.85, respectively. The values of the impulse response functions are in percent.

*Appendix C.2. Cap-and-Trade Regime*

The time-varying optimal environmental tax rate determined from the Ramsey problem is a useful benchmark to understand the level of the tax rate that optimizes the social welfare in different periods. However, such a frequent adjustment of the tax rate might not be feasible in reality when the cost of policy implementation is high. In this regard, we consider another environmental policy, namely, the cap-and-trade policy, and examine whether such a policy can result in labor dynamics and emission level that ensemble those resulted from the Ramsey optimal environmental tax rate.

A cap-and-trade requires the air pollutant emission level $Z_t$ to be fixed at a particular level $\bar{Z}$. The government in city $i$ then sells the permit to the intermediate good firms at a price $P_{i,Z,t}$. To have a fair comparison with the constant tax rate scenario presented in Section 4.3, $\bar{Z}$ is set to the deterministic steady-state value of the emission level when the environmental tax rate is 5%. For simplicity, we assume that both cities implement cap-and-trade policy (It is interesting to see if one city implements a cap-and-trade policy while the other implements an environmental tax policy, how would the labor and emission level respond differently to the shocks. We leave these topics for future research.).

Figures A6–A9 compare the different responses of labor and emission levels under the optimal environmental tax rate and the cap-and-trade regime to the four shocks. As shown, one distinctive

feature of the cap-and-trade regime is that the emission level remains constant and can completely insulate the impact of the shocks. Comparing the dashed lines in Figures 5 and A6, the responses of labor are quite similar under the constant tax rate and the cap-and-trade regimes: both of them produce more volatile responses of labor to those from the optimal environmental tax rate. In contrast, unlike Figure 6 that both the solid and dashed lines move closely together, Figure A7 shows that the labor dynamics are much more responsive under the cap-and-trade policy, revealing that the policy is unable to insulate the abatement inefficiency shock from the other city. An increase in abatement inefficiency in city $b$ is expected to drive up the permit price in the city. This would, in turn, raise the firms' operating cost and dampen the firms' production scale. Workers in city $b$ are forced to leave the city, leading to a large inflow of labor in city $a$ as shown in the figure.

As discussed in Section 4.3, the Ramsey optimal environmental tax rate produces dampened responses of labor to the shocks, compared to those generated from a constant tax rate. Surprisingly, as shown in Figures A8 and A9, the responses of labor are even less volatile than those produced by the Ramsey optimal environmental tax rate. This implies that simultaneous implementation of the cap-and-trade policies in both cities are capable of stabilizing the labor migration dynamics caused by the TFP shocks in both cities.

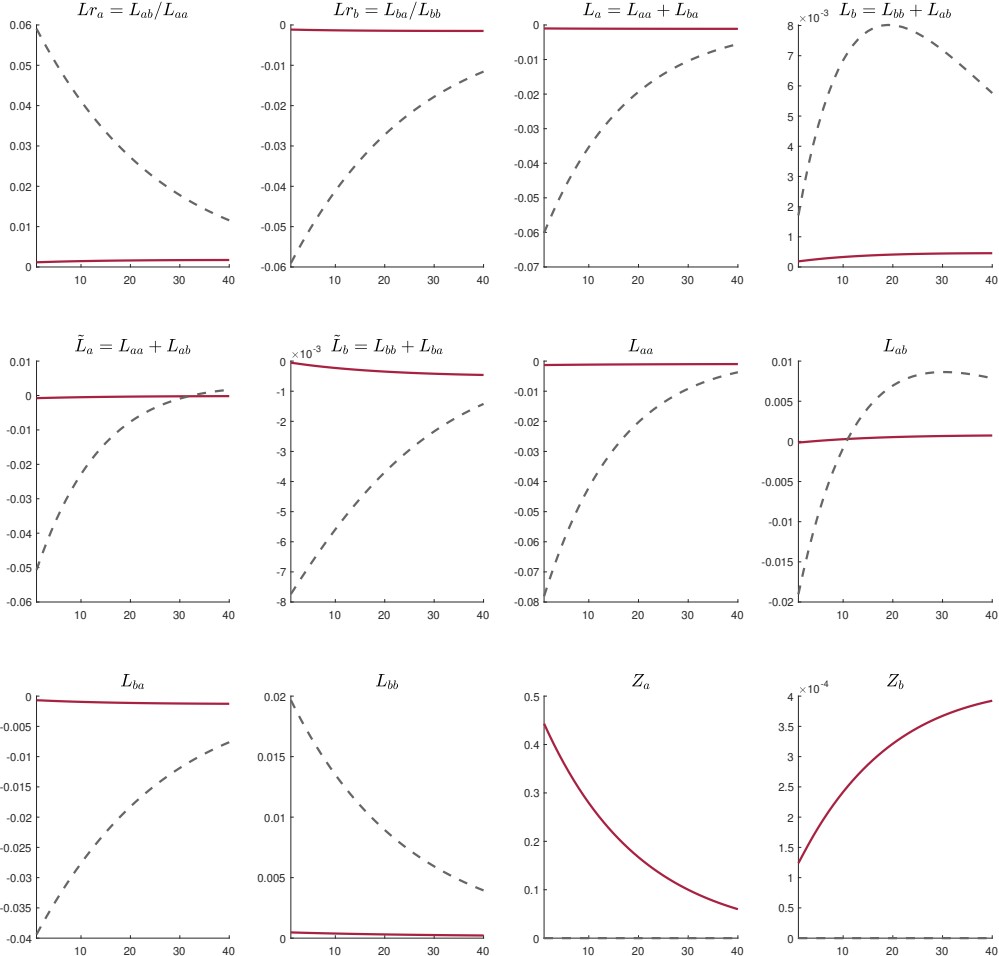

**Figure A6.** IRFs of labor and air pollutant emissions levels in both cities to one standard deviation of abatement inefficiency shock $\varphi_a$. The solid and dashed lines are the IRFs under the optimal tax rate and a cap-and-trade regime, respectively. The values of the impulse response functions are in percent.

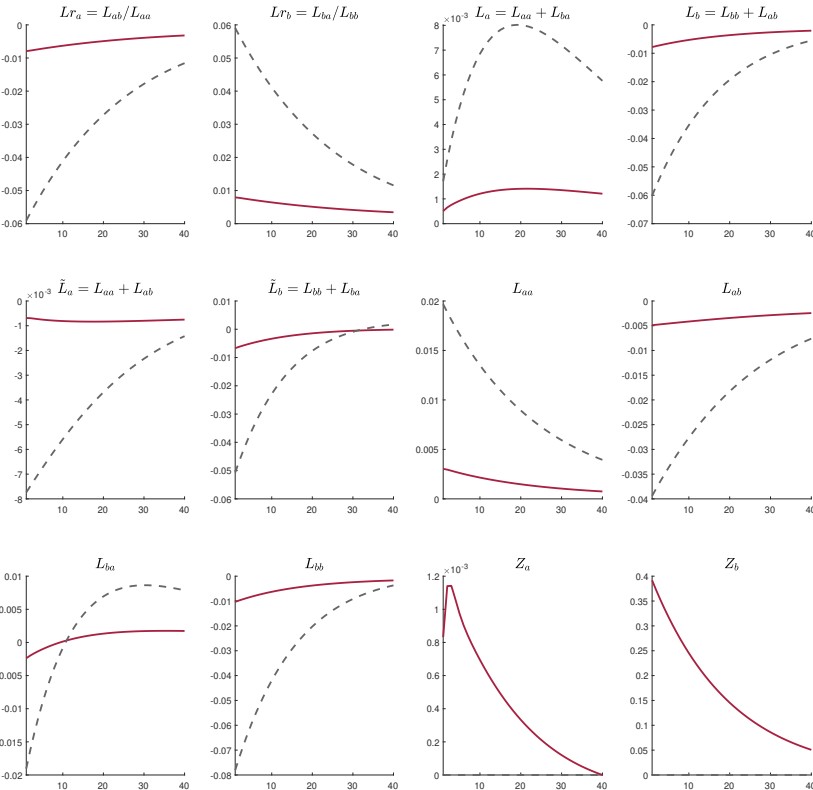

**Figure A7.** IRFs of labor and air pollutant emissions levels in both cities to one standard deviation of abatement inefficiency shock $\varphi_b$. The solid and dashed lines are the IRFs under the optimal tax rate and a cap-and-trade regime, respectively. The values of the impulse response functions are in percent.

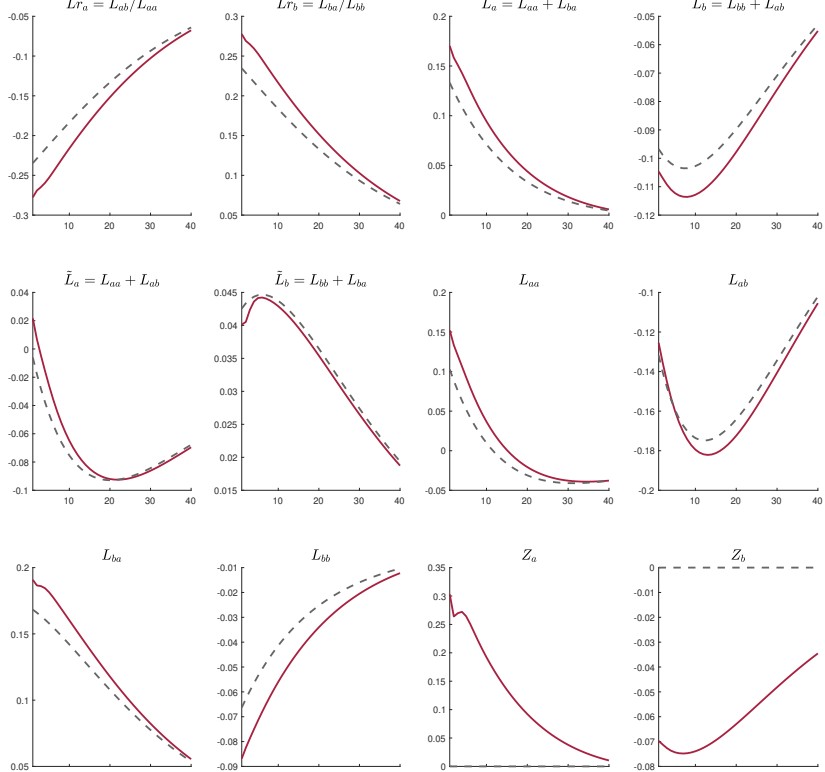

**Figure A8.** IRFs of labor and air pollutant emissions levels in both cities to one standard deviation of TFP shock $A_a$. The solid and dashed lines are the IRFs under the optimal tax rate and a cap-and-trade regime, respectively. The values of the impulse response functions are in percent.

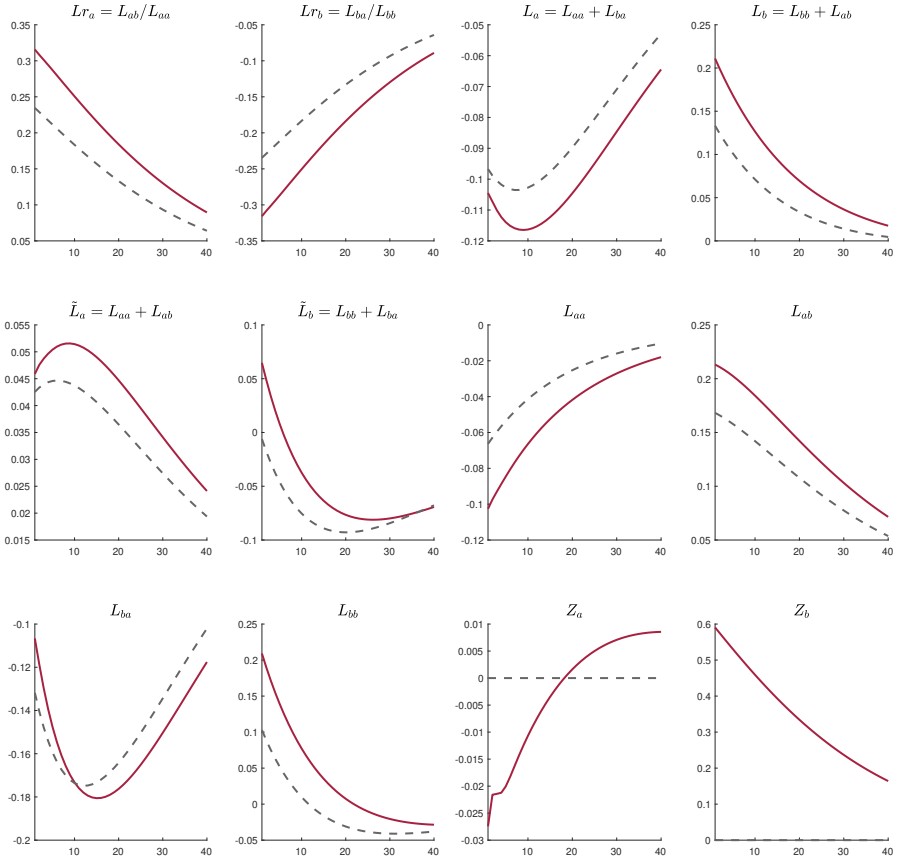

**Figure A9.** IRFs of labor and air pollutant emissions levels in both cities to one standard deviation of TFP shock $A_b$. The solid and dashed lines are the IRFs under the optimal tax rate and a cap-and-trade regime, respectively. The values of the impulse response functions are in percent.

*Appendix C.3. Non-Cooperative Optimal Environmental Tax Rates*

In Section 4.3, the optimal environmental tax rate is computed by assuming that the Ramsey social planner only appears in city $a$, while the environmental tax rate in city $b$ is assumed to be constant. One might wonder what is the optimal tax rate of city $a$ if there is also a Ramsey social planner in city $b$. In this setup, we solve for a pair of optimal environmental tax rates $\{p_{Z,a}, p_{Z,b}\}$ that constitutes a Nash equilibrium, in the sense that given the optimal tax rate $p_{Z,j}$, the Ramsey social planner in city $i$ find $p_{Z,i}$ that maximizes the lifetime utility of household in city $i$. Figure A10 compares the IRFs of the optimal environmental tax rate (of city $a$) under both scenarios. The solid lines are the same as those reported Figure 4, while the dashed lines are the resulting optimal environmental tax rate of the non-cooperative game. While the directions of movement for both paths are the same under all the four types of shocks, it is noting that the optimal environmental tax rates under the Nash equilibrium is less volatile. Compared to the solid lines where the environmental tax rate in city $b$ is constant, the same response of environmental tax rate in city $a$ becomes less effective if there is a counteracting measure proposed by the Ramsey social planner in city $b$. With a positive abatement inefficiency shock $\varphi_a$ or $\varphi_b$, the effect of reducing the environmental tax rate on attracting labor inflow would be canceled out by a similar measure adopted in the neighbor city. Without the benefits of labor inflow, it is no longer optimal to substantially reduce the tax rate. Hence, the Ramsey social planner mitigates the change in the tax rate so as to avoid triggering a dramatic policy response from another city. Similar logic applied in explaining the IRFs to the TFP shocks $A_a$ and $A_b$.

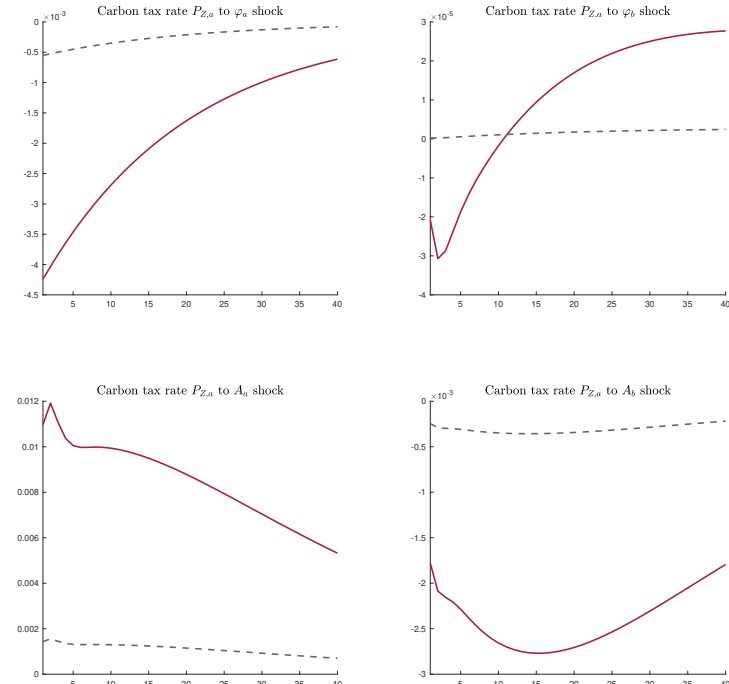

**Figure A10.** IRFs of the optimal environmental tax rate $p_{Z,a}$ to one standard deviation of abatement inefficiency $\varphi_i$ and TFP $A_i$ shocks in city $i \in \{a, b\}$. The solid lines are the IRFs where Ramsey social problem is solved only for city $a$, the dashed lines are the IRFs where Ramsey social problem are simultaneously solved for both cities.

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
