# Peer review of "Optimal Environmental Tax Rate in an Open Economy with Labor Migration—An E-DSGE Model Approach"

_sustainability, doi:10.3390/su11195147_

Round 1

Reviewer 1 Report

Article: Optimal Carbon Tax Rate in an Open Economy with Labor Migration – An E-DSGE Model Approach

Summary: The paper is interesting and does a good job of highlighting the importance of labor migration in the design of environmental policy. The paper is well written and the methodology is appropriate for research question addressed.

The principal issue that would prevent accepting the paper for publication in the journal Sustainability in its current form is conceptual in nature. The damages associated with carbon dioxide emissions are related to global emissions and location specific considerations (i.e. coastal risk of sea level rise, hurricane damage, etc). As greenhouse gas emissions are uniformly mixed in the atmosphere, the specific locations for emissions are not directly related to damages at that location and it is not appropriate to define a stock in each city.

Two possible modifications can address this issue:

Focus on air pollution rather than CO2 emissions. Allow the damage function and labor flows for one city (say a coastal city affected by sea level rise) to depend on emissions from both cities and damages for the other city to be less affected.

Likely the first modification is more in the spirit of the paper in its current form.

The results here are particularly interesting for smog, particulate matter and air pollution whereby policy efforts to improve air quality can attract workers and promote positive economic outcomes (depending on the ability to attract labor migration relative to the output costs of the tax). This argument is not valid for carbon dioxide and greenhouse gas emissions.

Minor notes:

There are multiple correlated externalities associated with fossil fuel combustion, including greenhouse gas emissions and air pollutant emissions. A link to this literature or a more explicit discussion of air pollution issues may be appropriate. Some good real world examples include the substantial air pollution in Beijing and in New Delhi and the implications of this pollution on migration decisions. The relationship between the study and the Kuznet’s curve can be made more explicit. The statement in the introduction that “environmental degradation is an inevitable outcome of economic development in a country’s early stage of development.” Is not well supported. Numerous studies suggest that the turning point suggested by the Kuznet curve does not materialize. The statement is, at best, debatable. The relationship between the Kuznet’s curve and the labor migration through the analysis of productivity shocks may be interesting. A substantial literature focuses on leakage from environmental policies across jurisdictions. That is, the introduction of stricter environmental policies in one location may shift economic activity to another and thereby not have as large as anticipated effect on global emissions. This is certainly a relevant concern for carbon dioxide emissions – less so for local air pollution – and appears to be somewhat supported by the model setup: “An increase in a city’s emissions would increase the emission level of the opponent city through the labor migration process.” These feedback mechanisms can be explored and related to the literature on leakage in environmental policy. A cap-and-trade system will have prices that adjust in a procyclical manner – low prices during contractions when demand is weak and high prices during periods of expansion. In practice, does this type of policy better match the centralized system intended with a tax? Do political constraints on frequent adjustments to the tax level and the extent to which these responses may be sticky or delayed in their approval/implementation suggest that a market based permit system might mirror better the optimal pricing policy for air pollutant emissions? Missing sentence at line 420

Reviewer 2 Report

The main objective of the paper is to analyze the role of labor flow in determining the optimal carbon tax rate. To do this, the author uses a two-city DSGE model that is an extension of Annicchiaro and Di Dio (2015) model.

COMMENTS

I found the paper seemingly hard to follow at times. In general, a clearer text is required.

[1] The interest of Annicchiaro and Di Dio (2015) lies in Business Cycle effects; so they use a New Keynesian model incorporating rigidities, capital adjustment costs, etc. However, this is not the case in this paper. I would recommend using a simpler model focused on the role of labor flow.

[2] The model should clarify the type of migration being referred to: migration between cities of the same country or migration between countries. This would affect the relationship between economies, the role of the government of each economy, the existence of a common government, etc.

[3] The model assumes only labor is allowed to move across cities and that capital does not flow freely. This is a very extreme assumption that completely conditions the results. Again it is important to clarify the kind of economies being referred to. An example could be helpful.

[4] The model assumes “emissions increase the labor disutility of working in the city, forces local workers to move out, and reduces the number of foreign workers that move in” (equations 2 and 3). Since this is central to the author´s analysis, it would be very useful to spend time motivating this slightly heroic assumption. The author should provide an adequate justification of this assumption and clarify the importance in explaining the results. 

[5] Assuming perfect mobility of labor implies assuming the presence of a single labor market condition. This will have consequences on wages in the two cities as well as on the aggregate resources constraint. None of this appears in the paper. 

[6] The description of the model is too concise, making it difficult to read. Recommendations to the Author include rewriting this section in line with Annicchiaro and Di Dio (2015). Furthermore, the model description contains errors. 

(7) Capital adjustment cost function is not specified. Also, the first order conditions seem to be a function of K (t) instead of K(t + 1).

[8] Condition (5) is wrong.  The general price level of city i at time t is missing. This would make conditions (7) and (8) wrong (see equation (18) in Annicchiaro and Di Dio, 2015). This is maintained in the Appendix so it looks like an error, not a typo. This restricts the results.

[9] Equation (6): R is not defined.

[10] Equation (9): q is the Tobin´s q but the authors do not mention it. In fact, this equation includes an undefined parameter.

[11] Equation (11): This equation is unclear. In Annicchiaro and Di Dio (2015) it is clearer. In addition, it is very important to clarify that firms producing final goods are symmetric and act under perfect competition and that the intermediate goods sector consists of a continuum of monopolistically competitive polluting producers.

[12] On page 7, line 231 quotes that “the higher the value of φ, the more inefficient the carbon abatement is”. Actually, φ measures emissions per unit of output in the absence of abatement effort.

[13] line 293, P* is not previously defined.

[14] The author should be more careful in presenting the calibrated values to avoid giving the impression that they have been arbitrarily chosen.  As an example, line 340 quotes “the steady-state value of emission stock, M, is around 800”, but it is unclear how it was obtained. In fact, some arbitrary values are assigned to several parameters. One must follow the usual practice of choosing parameters to reproduce some relevant characteristics of the economy or taking them directly from the literature, specifying the strategy employed in each case. In addition, how sensitive are the results to the choice of parameters? It would be interesting to see a sensitivity analysis in the paper.

[15] As the author points out in the conclusions, a strategic game between the two cities should be established using carbon taxes as a policy instrument. In my opinion, this could make a very interesting point in this paper.

Round 2

Reviewer 1 Report

The adjustments to the paper are well done and enhance the paper's appeal.

Reviewer 2 Report

I am satisfied with the changes made by the authors.